# PSPO: Trainable Potential-Based Reward Shaping with Internal Model Signals for Post-Training Policy Optimization of Large Language Models

## Abstract

Reinforcement learning from human feedback (RLHF) has become the de-facto paradigm for aligning large language models (LLMs), yet mainstream algorithms either incur the memory overhead of a value head (PPO) or remain vulnerable to sparse and miscalibrated rewards (GRPO, DPO). We propose **Potential-Shaped Policy Optimization (PSPO)**, a lightweight, *critic-free* framework that converts coarse scalar feedback into dense, context-aware signals by learning a trainable potential function. A 22.7M-parameter MiniLM network (the **Potential Network**) ingests inexpensive internal model signals (token embeddings, attention entropy, policy entropy) to produce adaptive shaping terms, while an alternating optimization scheme stably co-trains the policy and potential without extra rollouts. On eight English and Chinese mathematical-reasoning benchmarks, a Qwen2.5-14B model trained with PSPO achieves strong accuracy under a shared 300M-token RLHF budget (**68.1%** on GSM8K; **41.6%** on MATH) and exceeds PPO/DPO/GRPO by up to about 10 accuracy points across these benchmarks in this matched setting. Beyond math, PSPO also improves open-ended instruction following on ShareGPT and HelpfulQA under the same backbone. PSPO remains critic-free and adds only $< 3\%$ wall-clock overhead in our measurements, while yielding interpretable token-level reward attributions. Taken together, these results highlight signal-aware reward shaping as a practical route toward more efficient and stable RLHF for decoder-only language models in long-horizon, sparse-reward settings.

## 1 Introduction

*"The skilled strategist seeks victory from the momentum (*shi*), not from individuals." — Sun Tzu,* The Art of War*, "Shi".* Classical Chinese military theory emphasizes shaping the *configuration of forces* so that small local nudges produce a global advantage.

Classical Chinese military theory emphasizes shaping the *configuration of forces* so that small local nudges produce a global advantage. Here "momentum" is purely an analogy; throughout the paper $\Phi_\theta$ is a potential in the classical sense of potential-based reward shaping (Ng et al., 1999), not a value or $Q$-function. In post-training with reinforcement learning, however, we typically receive only a terminal score; this end-of-episode judgment carries little guidance for long chains of reasoning. Inspired by *shi*, we seek directional, stepwise signals along the trajectory: we train a lightweight *potential function* $\Phi_\theta$ on internal model signals of the policy (final-layer embeddings, attention entropy, and policy entropy) to estimate a state "potential," and we shape rewards by the difference $\gamma \, \Phi_\theta(s') - \Phi_\theta(s)$ to make feedback dense and context-aware while preserving optimal policies. In this view, $\Phi_\theta$ acts as a trajectory-level accumulator: it aggregates local signs of progress and releases them as dense, telescoping differences that preserve the optimum while accelerating credit assignment. We then co-train the policy and the Potential Network with an alternating-optimization scheme (AltOpt) and a trust region on successive $\Phi$ changes, stabilizing learning without extra rollouts. As we will show, this potential-based shaping yields consistent gains on mathematical reasoning benchmarks in English and Chinese at minimal overhead. Notably, the $< 3\%$ wall-clock

overhead refers to the alternating *potential phase* itself since both phases reuse the same rollouts; detailed measurements appear in Appendix H. This strategic perspective motivates our core design: a learnable potential that turns sparse terminal signals into dense, trajectory-aware credit without re-introducing a critic. While the mechanism is in principle applicable beyond math, all empirical evidence in this paper comes from a single decoder-only backbone (Qwen2.5-14B) and mainly math-focused RLHF settings, so we treat PSPO as a first step toward more signal-aware RLHF rather than a fully general solution.

**At-a-glance findings.** Across eight English/Chinese math benchmarks, PSPO consistently outperforms PPO (Schulman et al., 2017), DPO (Rafailov et al., 2024), and GRPO (Shao et al., 2024) under a matched 300M-token RLHF budget with $< 3\%$ additional wall-clock overhead (Sec. 5). Component ablations further show that potential shaping, internal model signals, and alternating optimization each contribute additively (Table 1). We always assume a standard scalar reward model and *shape rather than replace* its signal.

## 1.1 BACKGROUND AND MOTIVATION

RLHF pipelines typically expose only sparse, episode-level signals, which makes long-horizon credit assignment brittle–especially for chain-of-thought mathematics. Moreover, mainstream value-based methods add a memory-heavy value head, whereas critic-free methods (e.g., GRPO/DPO) often struggle when rewards are miscalibrated or delayed. We ask: can we densify feedback *without* reinstating a critic? Our answer is to learn a lightweight *potential function* from internal model diagnostics and to inject it via potential-based shaping, preserving optimal policies while accelerating credit assignment.

## 1.2 KEY INSIGHT

LLMs already expose rich internal model signals–token embeddings, attention entropy, policy entropy–that reflect uncertainty and reasoning progress. If a lightweight network learns a potential function over these signals, its difference $\gamma\Phi(s') - \Phi(s)$ can *densify* sparse rewards *without* re-introducing a costly critic, provided policy and potential are optimized in a mutually stable way.

## 1.3 OUR APPROACH: POTENTIAL-SHAPED POLICY OPTIMIZATION (PSPO)

PSPO is a critic-free RLHF framework that:

1. trains a 22.7M-parameter MiniLM to model the potential $\Phi_\theta$,
2. feeds internal model signals of the policy into $\Phi_\theta$ for fine-grained state estimation,
3. alternates between (i) updating the policy with shaped rewards and (ii) fitting $\Phi_\theta$ to return residuals, adding $< 3\%$ wall-clock overhead.

## 1.4 CONTRIBUTIONS

We summarize our main contributions as follows:

- **Dynamic reward shaping via a trainable potential function.** We introduce a lightweight, learnable **Potential Network** that dynamically augments sparse or miscalibrated rewards with $\gamma\Phi_\theta(s') - \Phi_\theta(s)$, preserving policy invariance while improving sample efficiency and stability.
- **Internal Model Signals for Potential Learning.** We leverage rich LLM diagnostics (final-layer token embeddings, attention entropy, and policy entropy) as inputs to the Potential Network, enabling fine-grained, context-aware reward adjustments without extra forward passes.
- **Alternating Optimization (AltOpt) of policy and potential.** We propose an AltOpt that interleaves policy updates on shaped returns with Potential Network updates on return residuals, yielding low-variance gradients and stable convergence with minimal wall-clock overhead.

Together, these results suggest that *signal-aware reward shaping* is a practical and scalable path to closing the reward sparsity gap in RLHF for decoder-only language models on long-horizon reasoning tasks under the regimes we study, and they highlight PSPO as a promising building block rather than a complete solution for all LLM alignment settings.

## 2 RELATED WORK

### 2.1 RLHF

RLHF has emerged as a key paradigm for aligning large language models (LLMs) with human intent (Ouyang et al., 2022; Glaese et al., 2022; Nakano et al., 2022). Traditional pipelines first train a reward model from human preferences, and then use policy optimization (typically PPO) to maximize the expected reward. While effective, this often relies on sparse, delayed scalar signals that lead to sample inefficiency and unstable convergence (Stiennon et al., 2022). Recent work has explored direct preference modeling to bypass reward fitting (Rafailov et al., 2024), but these methods may struggle with fine-grained temporal credit assignment.

### 2.2 REWARD SHAPING AND POTENTIAL-BASED METHODS

Reward shaping (Ng et al., 1999) introduces auxiliary signals to guide learning while preserving policy optimality. LM-Critic (Cao et al., 2024) produces token-level feedback via an external evaluator; PAR (Fu et al., 2025) frames shaping as bias correction with bounded priors. Dense Reward for Free (Chan et al., 2024) re-distributes a scalar reward along the trajectory using attention and gradient information, and on-policy distillation methods such as GKD (Agarwal et al., 2024) provide dense token-level learning signals by matching a teacher on self-generated rollouts. All of these inject dense signals but either rely on an extra critic/evaluator, treat the shaping function as *static*, or use a distillation-based objective rather than an explicit potential over internal diagnostics. PSPO instead learns a lightweight *potential* from *internal* policy signals and co-trains it with the policy under an explicit $\Phi$-trust-region, remaining critic-free and low-memory. In Appendix K we additionally provide a head-to-head comparison with LM-Critic, Dense Reward for Free, PAR, and an on-policy distillation baseline (GKD) under a common Qwen2.5-14B backbone and matched token budget, together with a complementary system-cost comparison (VRAM and wall-clock overhead).

### 2.3 INTERNAL MODEL SIGNALS FOR SHAPING

Most RLHF approaches treat LLMs as black boxes and use only external signals (reward score, KL, prompt metadata) for optimization (Bai et al., 2022). However, the internal activations of LLMs–such as token embeddings, attention distributions, and policy entropy–encode rich uncertainty and semantic information. LM-Critic (Cao et al., 2024) and ONI (Zheng et al., 2024) begin to explore such internal model signals, typically by using an external evaluator to generate token-level feedback. Our work takes a step further by directly feeding LLM-internal statistics into the Potential Network, enabling fine-grained, introspective reward adjustment with minimal computational overhead. Beyond ablations, Appendix B.3 and Appendix I describe an offline correlation analysis between these diagnostics and final correctness/reward (including partial correlations that regress out sequence length), as well as control experiments in which internal signals are randomized or shuffled or replaced by external-only features. These analyses are designed to test whether PSPO is exploiting non-trivial progress signals rather than trivial proxies such as response length.

### 2.4 ALTERNATING OPTIMIZATION AND CO-TRAINING

Joint optimization of policy and auxiliary modules (e.g., reward models or critics) is prone to instability due to non-stationary targets (Khetarpal et al., 2020). To address this, several works adopt AltOpt: RLAIF (Bai et al., 2022) freezes reward functions during policy updates. PSPO follows a cooperative two-phase optimization, alternating between fitting the potential function and updating the policy, which stabilizes gradients and avoids critic-induced variance while remaining fully critic-free.

### 2.5 POSITIONING OF PSPO

Compared to prior reward shaping methods that rely on static heuristics or external evaluators, PSPO introduces a lightweight, *trainable* Potential Network that leverages internal model states for shaping. Unlike PPO-based RLHF, it remains critic-free by design, and unlike DPO-based alignment (Rafailov et al., 2024), it supports dense token-level credit without explicit preference pairs.

Finally, its alternating optimization ensures low variance and efficient convergence, bridging the gap between reward-free and fully reward-dependent policy training. PSPO is therefore best viewed as a tool for long-horizon, sparse-reward settings—such as chain-of-thought math and instruction following where a single scalar reward must be propagated through many tokens and internal diagnostics correlate with "reasoning progress"—rather than a universal algorithm for all LLM training; in short, its effectiveness depends on the existence of informative internal signals and on tasks where temporal credit assignment is a dominant challenge.

## 3 Method

We introduce **Potential-Shaped Policy Optimization (PSPO)**, a reinforcement-learning framework that augments sparse or miscalibrated reward signals for LLMs with a *trainable* potential function. PSPO comprises three innovations:

1. **Trainable potential-based shaping** that adapts the reward at every step;
2. **Internal model signals** that feed rich LLM diagnostics into the Potential Network;
3. **AltOpt** that stably co-trains policy and potential.

The subsections below elaborate on each component, followed by the overall algorithm and a convergence sketch.

### 3.1 Problem Setup and Motivation

Consider an MDP with states $s$, actions $a$, transition $P$, and a reward model (possibly learned) that produces scalar feedback $r(s, a, s') \in [0, 1]$. In our RLHF experiments, the reward model emits a *single* scalar $R \in [0, 1]$ for each completed trajectory; for notational convenience we treat this as a per-step reward by *broadcasting* it along the sequence, i.e.,

$$r_t \equiv R \quad \text{for all } t,$$

and write $r(s_t, a_t, s_{t+1}) = r_t$ for the corresponding transition-level notation. This matches the usual bandit-style RLHF setup with a terminal score.

In RLHF for LLMs, such rewards are *sparse* and highly *variance-sensitive*. Classic potential-based shaping (Ng et al., 1999) adds a term $\gamma\Phi_\theta(s') - \Phi_\theta(s)$ with a *fixed* potential $\Phi$, preserving optimal policies but offering no adaptivity. Our goal is to endow this mechanism with *learnable*, adaptive shaping that reacts to evolving policy confidence while keeping the critic-free advantage of GRPO. We adopt a ratio-clipped policy objective (PPO-style clipping) without any value head; advantages are computed from normalized return-to-go (RTG), keeping the procedure critic-free while retaining the stability benefits of clipping. Here $\gamma \in [0, 1]$ is the discount factor, and $s'$ denotes the immediate successor state after emitting the current token.

**Assumed reward source & scope.** We assume a scalar reward $R$ provided by a standard RLHF reward model; PSPO *does not* introduce a new evaluator but shapes this reward via a learned potential (Eq. equation 1). Any systematic bias or miscalibration in $R$ can in principle be inherited or even amplified by the shaping term, and PSPO is *not* intended as a cure-all for reward-model bias. Instead, we view PSPO as complementary to ongoing efforts to improve reward models: it densifies whatever signal is available while controlling potential drift via an $\ell_\infty$ trust region. Appendix J and Appendix M further discuss this interaction and describe a small synthetic stress test with deliberately perturbed rewards to illustrate typical failure modes and partial safeguards.

### 3.2 Trainable Potential Function for Potential-Based Shaping

PSPO adopts classical potential-based shaping (Ng et al., 1999) but makes the potential *trainable*. We introduce a parametric potential $\Phi_\theta$ and define the shaped reward

$$r'(s, a, s') = r(s, a, s') + \gamma\Phi_\theta(s') - \Phi_\theta(s), \tag{1}$$

where $\theta$ is updated regularly (Sec. 3.4). This preserves policy invariance while providing an adaptive bias-correction term for sparse or miscalibrated rewards (formal analysis: Appendix A). Unless otherwise stated we use $\gamma = 1$ (bandit-style scalar rewards typical in RLHF); sensitivity to $\gamma$ is summarized in Appendix G.3.

### 3.3 POTENTIAL NETWORK: ARCHITECTURE AND INPUTS

We instantiate $\Phi_\theta$ as a lightweight encoder (MiniLM-L6-v2) that consumes a compact mix of *external* statistics (e.g., response log-likelihood, KL to a frozen reference, prompt tags) and *internal* diagnostics from the policy. Concretely, we use three inexpensive internal signals that summarize semantic progress and uncertainty; their *exact definitions*, discussion of *why* they help, and *feature variants* are deferred to Appendix B.3, and the corresponding *ablations* are reported in Appendix G.2. Because the Potential Network only maps a small set of aggregated diagnostics to a single scalar per state rather than predicting next tokens, this regression problem is much simpler than language modeling; an architecture ablation in Appendix G shows that a 22.7M-parameter MiniLM matches or nearly matches substantially larger potential architectures while being much cheaper. This keeps the main text focused while preserving reproducibility.

### 3.4 ALTERNATING OPTIMIZATION (ALTOPT) OF POLICY AND POTENTIAL

**Why alternating?** If one performs a *joint* gradient step on $(\psi, \theta)$, the non-stationary reward $r'$ causes biased policy gradients (Khetarpal et al., 2020); most shaping papers therefore **freeze** $\Phi$ entirely. Our solution sits between these extremes.

**Mechanism.** We adopt **Alternating Optimization (AltOpt)** with two phases (see Appendix B for pseudocode): *(i) Policy phase* fixes $\theta$ and updates $\psi$ on shaped returns; *(ii) Potential Network phase* fixes $\psi$ and fits $\theta$ to a return target (see Appendix B).

**Mutual stability.** Intuitively, we want the potential to change slowly compared to the policy. When successive changes of $\Phi_\theta$ are bounded and the policy update obeys a standard trust region (e.g., a PPO-style KL bound), the shaped and unshaped $Q$-functions induce the same optimal policies while the shaping term acts as a low-variance baseline. A formal statement and verification of these assumptions (including the precise definition of the potential drift $\delta_k$) are given in Appendix A.

**Benefits.**

1. **Stable credit assignment**: each phase sees a quasi-stationary objective.
2. **Lower variance**: $\Phi_\theta$ quickly tracks long-horizon returns, reducing advantage variance.
3. **No extra rollout**: both phases reuse the *same* batch, adding $< 3\%$ wall-clock overhead in our setup.

### 3.5 OPTIMIZATION OBJECTIVES FOR PSPO

We now make the training objectives explicit. Recall that in our RLHF setup the reward model emits a scalar $R$ per trajectory and we broadcast it along the sequence, so that $r_t \equiv R$ for all $t$. Given a potential $\Phi_\theta$, we form *per-token* shaped rewards

$$r'_t \; = \; r_t \; + \; \gamma\, \Phi_\theta(s_{t+1}) \; - \; \Phi_\theta(s_t), \tag{2}$$

and define the token-level return-to-go (RTG) from step $t$ as

$$G_t \; = \; \sum_{u=t}^{T} \gamma^{u-t}\, r'_u, \tag{3}$$

where $T$ is the last token index of the trajectory. We then compute *normalized shaped advantages*

$$A_t^{\text{shape}} \; = \; \frac{G_t - \mu_G}{\sigma_G}, \tag{4}$$

where $\mu_G$ and $\sigma_G$ are the mean and standard deviation of $\{G_t\}$ within the current minibatch. No value head is trained; PSPO remains critic-free and uses these advantages directly.

**Policy phase.** Fixing $\theta$, we update the policy by minimizing a PPO/GRPO-style clipped loss

$$\mathcal{L}_{\text{PPO}} = -\,\mathbb{E}\Big[\min\big(\rho_t(\psi)\, A_t^{\text{shape}}, \text{clip}\big(\rho_t(\psi),\, 1-\varepsilon,\, 1+\varepsilon\big)\, A_t^{\text{shape}}\big)\Big], \tag{5}$$

where

$$\rho_t(\psi) \;=\; \frac{\pi_\psi(a_t \mid s_t)}{\pi_{\psi^-}(a_t \mid s_t)}$$

is the usual probability ratio and $\varepsilon > 0$ is the clipping parameter. We use a frozen reference $\pi_{\psi^-}$ for KL control, but do not train any value head.

**Potential phase.** Fixing $\psi$, we temporarily *turn off* shaping by setting $\Phi_\theta \equiv 0$ and compute an *unshaped* RTG

$$\hat{G}_t \;=\; \sum_{u=t}^{T} \gamma^{u-t} r_u, \tag{6}$$

from the original rewards $r_u$ (which reduce to multiples of $R$ in the single-terminal-reward case). We normalize these returns within the minibatch to obtain an *unshaped, centered advantage*

$$\hat{A}_t \;=\; \frac{\hat{G}_t - \mu_{\hat{G}}}{\sigma_{\hat{G}}}, \tag{7}$$

with $\mu_{\hat{G}}$ and $\sigma_{\hat{G}}$ the mean and standard deviation of $\{\hat{G}_t\}$. In the single-terminal-reward regime used in our RLHF experiments, broadcasting $R$ implies that all $r_u$ share the same scalar, so $\hat{G}_t$ is simply the (discounted) future sum of this scalar; we nevertheless treat $\hat{A}_t$ as a token-level target indexed by $t$, which matches standard critic-free GRPO-style implementations and keeps the notation compatible with general multi-step reward settings. When we later refer to "unshaped advantages with a small trust region", we mean precisely the quantities $\hat{A}_t$ together with the regularized objective $\mathcal{L}_{\text{shape}} + \mathcal{L}_{\text{TR}}$ in Eqs. equation 8–equation 9.

The Potential Network is then trained to fit its per-step difference to this target:

$$\mathcal{L}_{\text{shape}} = \mathbb{E}_{(s_t, s_{t+1})}\Big[\big(\gamma\Phi_\theta(s_{t+1}) - \Phi_\theta(s_t) - \hat{A}_t\big)^2\Big] \;+\; \mathcal{L}_{\text{TR}}(\theta), \tag{8}$$

where $\mathcal{L}_{\text{TR}}$ is a small trust-region penalty on successive potentials. Concretely, in all experiments we use

$$\mathcal{L}_{\text{TR}}(\theta) \;=\; \lambda_{\text{tr}} \mathbb{E}_{s \sim \mathcal{B}}\Big[\big(\Phi_\theta(s) - \Phi_{\theta^-}(s)\big)^2\Big], \tag{9}$$

where $\theta^-$ denotes the previous-iteration snapshot and $\mathcal{B}$ is the current minibatch. This soft trust region directly controls the successive potential drift $\delta_k = \max_s |\Phi_{\theta_{k+1}}(s) - \Phi_{\theta_k}(s)|$ and implements the bounded-drift assumption used in Proposition A.1. In Appendix B we summarize the combined objective as $\mathcal{L}_{\text{shape}} + \mathcal{L}_{\text{TR}}$ (see Eq. equation 17).

This explicit decoupling—shaped advantages for the policy and unshaped, centered advantages for the potential—avoids self-coupling between the shaping term and its training target and reduces variance (Prop. B.1).

## 3.6 Algorithm and Workflow

The overall training loop alternates between a *policy phase* (PPO on shaped advantages) and a *Potential Network phase* (fit $\Phi_\theta$ to an unshaped advantage with a small trust region), reusing the same batch. See Appendix B for the full pseudocode. Below we keep a brief text workflow and the overview figure.

**Workflow (text).** (1) Generate rollouts, pass each completed trajectory through the reward model to obtain a scalar score $R$, and in parallel compute a KL penalty between the current policy and a frozen reference model; (2) broadcast $R$ along the sequence to form per-token rewards $r_t$ and evaluate $\Phi_\theta(s_t), \Phi_\theta(s_{t+1})$ in order to compute shaped rewards $r'_t = r_t + \gamma\Phi_\theta(s_{t+1}) - \Phi_\theta(s_t)$; no value head is used. We form token-level return-to-go (RTG) from $r'_t$ and normalize it within the batch to obtain $A^{\text{shape}}$, while the KL term from the reference model enters only the policy loss as a separate regularizer. (3) *Policy phase:* update $\psi$ with ratio clipping on $A^{\text{shape}}$ using the critic-free PPO-style objective in Eq. equation 5; (4) *Potential Network phase:* fix $\psi$ and fit $\theta$ so that $\gamma\Phi_\theta(s_{t+1}) - \Phi_\theta(s_t)$ matches the unshaped, centered target $\hat{A}_t$ under a small trust region (Eqs. equation 7–equation 8 and equation 9); (5) reuse the same batch for both phases and alternate the two updates.

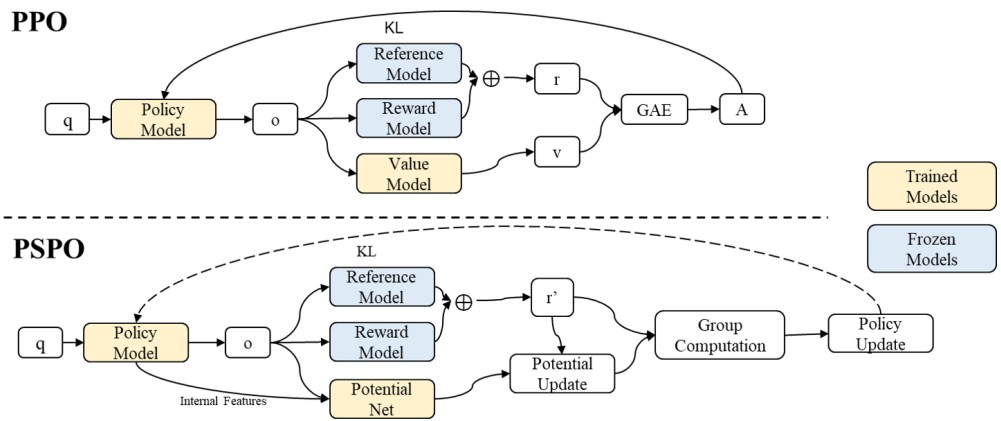

Figure 1: **Overview of PSPO training workflow.** The reward model produces a scalar trajectory-level score $R$, and a separate KL penalty to a frozen reference model is added to the policy loss; in the diagram these two contributions are grouped into a single "RM + KL" block for visual simplicity only. Details, full pseudocode, and extended diagrams are provided in Appendix B and Appendix I.

## 4 THEORETICAL ANALYSIS

We give only a high-level summary here and defer formal statements and proofs to Appendix A.

At a high level, we show that PSPO's learned potential $\Phi_\theta$ behaves like a low-variance, critic-free baseline that *does not* change the optimal policy under a mild bounded-drift condition. Concretely:

- We extend classical potential-based shaping (Ng et al., 1999) to the time-varying case and prove a *policy-invariance* result: as long as successive changes of $\Phi_\theta$ are bounded (enforced by the $\ell_\infty$ trust region in Sec. 3.5), the shaped and unshaped MDPs share the same optimal policies.

- We compare targets for training the Potential Network and show that using an *advantage-based* target (unshaped, centered return) reduces variance compared to fitting raw Monte Carlo returns while keeping the estimator unbiased for the desired shaping term.

- We verify that the AdamW + cosine decay schedule and trust-region regularization used in our experiments satisfy the bounded-drift assumptions, and we provide explicit bounds on the induced shaping bias relative to the optimal action gap.

These results justify using a learned potential as a critic-free, low-variance shaping mechanism without altering the set of optimal policies in the regimes we study; full details are in Appendix A.

## 5 EMPIRICAL EVALUATION

### 5.1 EXPERIMENT SETUP

We summarize the setup briefly and defer full protocols to Appendix E. We start from the publicly released **Qwen2.5-14B** *base* checkpoint (not the stronger instruct or RL-tuned variants) and keep the supervised fine-tuning stage fixed across all methods. All RLHF variants (PPO, DPO, GRPO, PSPO and dense baselines) are trained under the same **300M generated-token budget** on the same mixture of English/Chinese math prompts; the exact data sources, filtering rules, and reward model details are listed in Appendix E.

Our goal in this paper is *not* to match proprietary Qwen2.5-instruct or vendor-level GSM8K numbers (which exceed 90% and rely on substantially larger, closed training pipelines (Qwen et al., 2025)), but to study a controlled, mid-budget RLHF regime on a shared backbone. Restricting to public math RLHF data and a 300M-token budget keeps all methods on equal footing and makes it possible to

| Variant | English Benchmarks | | | | | Chinese Benchmarks | | |
|---|---|---|---|---|---|---|---|---|
| | GSM8K | MATH | OCW | SAT | MMLU$_{STEM}$ | CMATH | GaokaoMathCloze | GaokaoMathQA |
| PPO | 54.1 | 32.5 | 8.2 | 76.1 | 48.8 | 71.2 | 18.9 | 37.7 |
| PPO+DenseCritic | 61.0 | 36.0 | 12.4 | 79.5 | 53.3 | 72.7 | 19.7 | 38.9 |
| PPO+ShapedReward | 66.4 | 39.7 | 16.2 | 88.0 | 58.1 | 76.8 | 24.3 | 41.0 |
| DPO | 57.0 | 33.2 | 9.1 | 77.7 | 46.9 | 72.5 | 18.8 | 37.5 |
| GRPO | 65.6 | 37.3 | 15.4 | 80.6 | 57.6 | 74.1 | 20.3 | 39.8 |
| PSPO (–Shaping) | 65.1 | 38.2 | 15.4 | 80.1 | 57.7 | 74.5 | 21.0 | 39.9 |
| PSPO (–InternalSignals) | 66.4 | 38.6 | 16.1 | 88.7 | 57.7 | 75.2 | 21.0 | 40.5 |
| PSPO (–AltOpt) | 66.1 | 38.1 | 16.3 | 88.3 | 57.8 | 76.0 | 21.1 | 40.1 |
| **PSPO (full)** | **68.1** | **41.6** | **18.4** | **90.2** | **59.0** | **78.8** | **27.1** | **42.3** |

Table 1: Main results and ablations (%). Upper block: baselines; lower block: component ablations. **PSPO (full)** performs best overall. Removing any single component leads to statistically significant drops ($p < 0.01$, paired $t$-test). Underline = second best; bold = best. All scores are mean over 3 seeds; all other settings are held constant across variants.

attribute performance differences to the optimization algorithm rather than to data scale or extra pretraining.[1]

To make convergence behavior comparable, Appendix H further reports accuracy-vs-steps and reward-vs-steps curves on GSM8K and MATH (Fig. 3), showing that all methods have largely plateaued under this budget and that PSPO maintains its advantage throughout training.

We use **Qwen2.5-14B** as the base policy and a lightweight **MiniLM-L6-v2** Potential Network; training alternates between the policy and Potential Network phases on the same rollouts. Hardware, batch schedules, optimization hyperparameters, shaping/KL settings, token budgets, and software stack: Appendix E. **Unless otherwise specified, decoding (temperature 0.7, top-$p$ 0.9, max 512) and KL control ($\beta{=}0.04$ to a frozen reference) are kept identical across all methods.**

**Baseline sanity check.** We verify that our PPO and GRPO implementations match open-source baselines on smaller models; detailed numbers are reported in Appendix E (Table 4). This sanity check suggests that the performance differences in Sec. 5 are not due to implementation errors.

## 5.2 MAIN RESULTS AND ABLATIONS

*Notation for ablations.* PSPO (–Shaping): remove potential-based shaping ($\Phi_\theta \equiv 0$); PSPO (–InternalSignals): disable internal model signals, keep only external stats; PSPO (–AltOpt): replace alternating optimization with joint updates. When we set $\Phi_\theta \equiv 0$, the shaped rewards $r'_t$ collapse back to the broadcasted scalar reward and the policy objective reduces to the same critic-free, normalized-RTG + clipping scheme as in GRPO; PSPO (–Shaping) can therefore be viewed as our in-framework reproduction of GRPO, differing only in minor implementation choices (e.g., normalization and batching). Variants such as Dr.GRPO further modify clipping and normalization schedules but still operate without a value head; we position PSPO as complementary, focusing on how to *densify* the underlying scalar signal.

### 5.2.1 DATASETS AND TASKS

We evaluate PSPO and all baselines on a mixture of English and Chinese mathematical-reasoning benchmarks (GSM8K (Cobbe et al., 2021), MATH (Hendrycks et al., 2021), OCW (Lewkowycz et al., 2022), SAT (Azerbayev et al., 2024), MMLU-STEM (Wang et al., 2024), CMATH (Wei et al., 2023), GaokaoMathCloze (Zhong et al., 2023), GaokaoMathQA (Zhong et al., 2023)) and two open-ended instruction-following sets (ShareGPT, HelpfulQA). These cover both structured chain-of-thought math and general instruction following. Detailed dataset descriptions, splits, and licenses are deferred to Appendix E and Appendix F.

---

[1]Absolute accuracies in our tables are therefore not directly comparable to the Qwen2.5 technical report or to FR-EEE / GHPO results trained with much larger and partly proprietary data.

### 5.2.2 BASELINES AND CONTROLS

To evaluate PSPO, we compare against several representative and competitive baselines and two strengthened PPO variants that explicitly control for the effect of dense rewards:

- **PPO** (Schulman et al., 2017): Standard RLHF baseline with a separate value head and *sparse, episode-level* reward.
- **PPO+DenseCritic**: PPO equipped with *token-wise value estimation* (dense critic head) trained on the *same* dense shaped reward used by PSPO, isolating the benefit of dense credit assignment.
- **PPO+ShapedReward**: PPO baseline that augments the sparse scalar reward with a dense potential-shaped reward $r'$ from Eq. equation 1 (computed using the same Potential Network as PSPO), while still relying on its own value head for advantage estimation.
- **DPO** (Rafailov et al., 2024): Direct preference optimization without online rollouts.
- **GRPO** (Shao et al., 2024): critic-free optimization with constant clipping.

All methods use the same base model (**Qwen2.5-14B**) and are trained with identical compute budgets (300M generated tokens on $2 \times$A800 GPUs). Results are averaged over 3 seeds to ensure statistical reliability.

### 5.2.3 PERFORMANCE COMPARISON

PSPO consistently outperforms PPO, DPO and GRPO across all datasets, with largest gains on MATH and CMATH. This highlights the unique advantage of trainable, signal-aware shaping over value-based or preference-based strategies.

Table 1 reports accuracy across all tasks. PSPO outperforms all baselines on most benchmarks, especially those with sparse or misaligned rewards (e.g., MATH, CMATH), indicating the effectiveness of dynamic reward shaping.

**Observations:**

- **Early-stage stability:** PSPO improves GSM8K and SAT scores due to enhanced early shaping and policy regularization.
- **Hard-reward domains:** On MATH and CMATH, PSPO achieves notable gains, suggesting that the **Potential Network** helps mitigate sparse rewards.
- **Cross-lingual generalization:** PSPO generalizes well across English and Chinese tasks without any language-specific tuning.

Compared to PPO+DenseCritic trained on the same dense targets, PSPO remains critic-free and attains competitive or better accuracy with lower memory footprint, suggesting the learned potential is a lightweight substitute for dense value heads.

### 5.2.4 OPEN-ENDED GENERATION

Detailed results on ShareGPT / HelpfulQA, including metrics, judging protocol, and analysis, are provided in Appendix F.

## 5.3 ABLATION STUDY

*All ablation experiments are in Appendix G.*

### 5.3.1 TRAINING EFFICIENCY AND RESOURCE ANALYSIS

For training speed, VRAM usage, tokens-per-dollar, energy estimates and parallel efficiency, see Appendix H.

### 5.3.2 FINE-GRAINED REWARD ATTRIBUTION ANALYSIS

Detailed distributional comparisons of raw vs. shaped rewards, temporal dynamics of summary statistics, and token-level potential heatmaps are provided in Appendix I.

## 6 CONCLUSION

### 6.1 SUMMARY OF FINDINGS.

We presented PSPO, a lightweight and critic-free reinforcement learning framework that addresses the challenge of sparse and misaligned rewards in LLM fine-tuning. By leveraging adaptive potential-based shaping, internal model signals, and alternating optimization, PSPO introduces a scalable, introspective approach to credit assignment.

Empirical results on eight English and Chinese math benchmarks confirm that PSPO significantly outperforms PPO, DPO, and GRPO, particularly on tasks with delayed or sparse feedback. Ablation studies verify that each component–Potential Network, internal model signals, and alternating training–plays a critical and complementary role.

These findings highlight that **adaptive reward shaping based on internal model diagnostics can substantially improve RLHF performance** in the long-horizon, sparse-reward regimes we study, bridging key gaps in temporal credit assignment and model alignment. PSPO thus provides a practical path toward more efficient and stable reinforcement learning for decoder-only language models on long-horizon reasoning tasks, rather than a universal recipe for all LLM alignment settings, while incurring only minimal overhead.

### 6.2 LIMITATIONS

We note four limitations and practical mitigations.

1. **Dependence on the reward model.** PSPO inherits biases and miscalibration from the underlying scalar reward and cannot, by itself, correct a systematically wrong reward model. *Mitigations:* periodic red-teaming of the reward model, ensembling heterogeneous RMs, and capping successive potential drift via the $\ell_\infty$ trust region to avoid runaway shaping (Appendix J). Appendix M outlines a small synthetic stress test with perturbed rewards (e.g., length-biased or saturated scores) designed to probe how PSPO and GRPO behave under controlled miscalibration; a complete treatment of adversarial or highly biased reward models is left for future work.
2. **Lack of explicit interpretability constraints for the Potential Network.** $\Phi_\theta$ is trained on scalar returns without token-wise priors, which may limit attribution reliability and transfer beyond math. *Mitigations:* add sparsity/stability regularizers, probe-based constraints, or post-hoc audits on token-level attributions.
3. **Scope of evaluations.** Experiments target English/Chinese math and a small set of instruction-following tasks; broader languages, modalities, and subjective alignment remain open. *Mitigations:* extend to multilingual/multimodal settings and incorporate human-in-the-loop assessments.
4. **Single backbone and architecture dependence.** All empirical results use a single decoder-only backbone (Qwen2.5-14B) and a single RLHF stack. Internal diagnostics such as attention entropy and token embeddings may behave differently for other architectures (e.g., Mixture-of-Experts, encoder–decoder models). *Mitigations:* replicate PSPO on alternative backbones (e.g., LLaMA-3, Mistral, DeepSeek-style models) and characterize when internal signals remain informative and when they fail.

For extended discussion of security/privacy (internal-signal leakage), potential bias amplification, and compute/energy footprint, see Appendix J.

**Artifacts.** We plan to release training and evaluation scripts, configuration files (including all seeds), and a reproducibility README; dataset and model licenses are enumerated in Appendix E.

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

# A  THEORETICAL ANALYSIS (FULL VERSION)

**Background and relation to successor features.**  Classic potential-based shaping (Ng et al., 1999) can be viewed through the lens of the *successor representation* (Dayan, 1993), in which a potential function approximates the discounted future occupancy of successor states. Our trainable $\Phi_\theta$ generalizes this idea to high-dimensional language settings and is closely related to *successor features* (Barreto et al., 2018), but replaces handcrafted basis vectors with token-level diagnostics extracted from the LLM itself. The shaped reward $r' = r + \gamma \Phi_\theta(s') - \Phi_\theta(s)$ therefore inherits the theoretical guarantees of successor-feature methods while remaining critic-free and lightweight.

**Adaptation view.**  Classic successor features (SF) enable fast adaptation by decoupling dynamics from rewards. PSPO's potential can be viewed as a learned regressor of discounted *diagnostic* accumulators, so when the reward model or prompt mix shifts, updating $\Phi_\theta$ re-aligns dense shaping without introducing a value head. This yields *intra-task* fast re-calibration (policy fixed within the potential phase), complementary to SF-style *inter-task* transfer.

**Why learn $\Phi$ instead of fixing it?**  (i) *Adaptive*: $\Phi_\theta$ tracks the evolving state distribution and mitigates miscalibrated or delayed rewards; (ii) *Stable and memory-light*: alternating optimization plus a trust region on successive $\Phi$ reduces advantage variance without adding a critic; (iii) *Cheap*: both policy and potential phases reuse the same rollouts, so the measured overhead is below $3\%$ in our setup.

**Notation.**  We write $H$ for the (finite) episode horizon (number of time steps). To avoid collisions with attention notation, we denote the number of attention heads by $N_{\text{head}}$ and the number of Transformer layers by $L$. $\Delta_\star(s) = Q^\star(s, a^\star) - \max_{a \neq a^\star} Q^\star(s, a)$ is the optimal action gap. $\delta_k = \max_s |\Phi_{\theta_{k+1}}(s) - \Phi_{\theta_k}(s)|$ measures the successive potential change. We control the drift of $\Phi$ by a trust region with radius $\tau_k$: $\|\Phi_{\theta_{k+1}} - \Phi_{\theta_k}\|_\infty \leq \tau_k$ (cf. Eq. equation 15). The token-level shaped and unshaped returns $G_t, \hat{G}_t$ and advantages $A_t^{\text{shape}}, \hat{A}_t$ are as in Sec. 3.5. Unless stated, bandit-style rewards use $\gamma = 1$; discounted variants are given for completeness. Here $\tau_k$ denotes a per-iteration $\ell_\infty$ trust-region radius limiting the successive change of the potential.

Potential-based shaping is known to preserve optimal policies when the potential function is fixed: Ng et al. (1999) show that adding $F(s_t, s_{t+1}) = \gamma \Phi(s_{t+1}) - \Phi(s_t)$ to the reward leaves the optimal policy set unchanged. We restate this result for slowly updated potentials produced by PSPO.

**Proposition A.1** (Policy invariance under bounded cumulative drift; cf. Ng et al. (1999)). *Assume rewards and potentials are uniformly bounded and that the sequence $\{\Phi_{\theta_k}\}$ has bounded cumulative drift in the above sense. Then the shaped process with rewards $r'_t = r_t + \gamma \Phi_{\theta_k}(s_{t+1}) - \Phi_{\theta_k}(s_t)$ shares the same optimal policies as the original MDP with rewards $r_t$. Intuitively, the shaping term still induces a telescoping bias whose discounted cumulative effect is uniformly bounded and therefore cannot change action orderings when the action gap is sufficiently large.*

**Corollary A.2** (Gap-protected optimality). *If the discounted future drift is below a constant fraction of the optimal action gap, shaping cannot flip the argmax.*

## A.1  ASSUMPTION VERIFICATION (SUMMARY)

We now make the "bounded cumulative drift" assumption concrete for the *actual* optimizer and trust region used in PSPO. Let the Potential Network parameters be updated by AdamW with cosine decay (exactly as in Appendix E):

$$\theta_{k+1} = \theta_k - \eta_k \hat{g}_k - \eta_k \lambda_w \theta_k, \qquad \eta_k = \frac{\eta_0}{2}\left(1 + \cos(\frac{\pi k}{T})\right), \tag{10}$$

where $\hat{g}_k$ is the bias-corrected gradient, $\lambda_w$ is weight decay, and $T$ is the training horizon. Assume: (i) $\|\hat{g}_k\| \leq G$, (ii) $\|\theta_k\| \leq B$ (e.g., via clipping/projection), and (iii) $\Phi_\theta$ is $L$-Lipschitz in $\theta$. Then the successive potential change

$$\delta_k := \max_s \left|\Phi_{\theta_{k+1}}(s) - \Phi_{\theta_k}(s)\right| \leq L \eta_k (G + \lambda_w B), \tag{11}$$

and hence $\sum_{k=0}^{T-1} \delta_k \leq L(G + \lambda_w B)\frac{\eta_0 T}{2} < \infty$. Therefore the discounted (or finite-horizon, when $\gamma = 1$) cumulative shaping bias is finite, and the policy-invariance conclusions of Proposition A.1 apply.

In addition to this analytic bound, we empirically monitor $\delta_k$ throughout PSPO training (Appendix I; see Fig. 4 and Table 11): its empirical envelope decays over time and stabilizes at a small fraction of the observed advantage gap, and the trust-region projection in Eq. equation 15 is only activated for a small fraction of batches early in training. This directly ties the mathematical assumption $\sum_k \delta_k < \infty$ to both the concrete update rule in Algorithm 1 and to the measured training dynamics.

**Mutual stability (formal).** We say that the policy–potential pair $(\pi_\psi, \Phi_\theta)$ is *mutually stable* if along the training trajectory: (i) the per-iteration potential drift

$$\delta_k \;:=\; \max_s \big|\Phi_{\theta_{k+1}}(s) - \Phi_{\theta_k}(s)\big|$$

is summable, i.e. $\sum_{k=0}^{\infty} \delta_k < \infty$, and (ii) the policy update satisfies a standard trust region (for example a PPO-style KL bound between successive policies). Under these conditions the shaped and unshaped $Q$-functions differ by a uniformly bounded telescoping term and therefore induce the same optimal policy whenever the action gap is non-zero (Proposition A.1). The concrete trust-region mechanism and its verification under AdamW+cosine decay are given in Eqs. equation 15–equation 17 and Appendix D.

**Explicit bounds in the two regimes.** For clarity we make the two cases explicit:

$$B_t^{\text{finite}}(H) \;:=\; \sum_{j=t}^{t+H-1} \delta_j, \tag{12}$$

for the finite-horizon bandit-style setting ($\gamma{=}1$, episode length $H$), and

$$B_t^{\text{disc}}(\gamma) \;:=\; \sum_{j=t}^{\infty} \gamma^{j-t} \delta_j \;\leq\; \frac{1}{1-\gamma}\,\bar{\delta}_{t:\infty}, \tag{13}$$

for the discounted case $\gamma \in (0,1)$, where $\bar{\delta}_{t:\infty} \triangleq \sup_{j\geq t} \delta_j$ (or any admissible envelope). Both $B_t^{\text{finite}}$ and $B_t^{\text{disc}}$ upper-bound the telescoping bias induced by shaping; when these bounds are small relative to the optimal action gap, Proposition A.1 guarantees action-order preservation.

*Rule of thumb.* Mapping $(1-\gamma)^{-1}$ to an "effective horizon" $H$ is a coarse heuristic for comparing the two regimes; the strict guarantees follow Eqs. equation 12–equation 13 and Proposition A.1.

Two common variants also satisfy bounded drift:

- **Robbins–Monro steps**: if $\eta_k = a/(b+k)^p$ with $p \in (1/2, 1]$, then $\sum_k \eta_k < \infty \Rightarrow \sum_k \delta_k < \infty$.
- **Trust region on** $\Phi$: enforcing $\|\Phi_{\theta_{k+1}} - \Phi_{\theta_k}\|_\infty \leq \tau_k$ with $\sum_k \tau_k < \infty$ directly yields $\sum_k \delta_k < \infty$.

Full derivations and practical tuning rules are deferred to Appendix D. In addition to this analytic bound, we empirically monitor $\delta_k$ throughout PSPO training (Appendix I): its empirical envelope decays over time and stabilizes at a small fraction of the observed advantage gap, and the trust-region projection is only activated for a small fraction of batches early in training, supporting the bounded-drift assumption in practice.

# B FULL OBJECTIVES AND ALGORITHM

## B.1 OPTIMIZATION OBJECTIVES FOR PSPO

Having outlined the rationale and mechanism for alternating optimization, we now detail the specific objectives and losses used in each phase of PSPO training. The basic shaped reward $r'_t$, return-to-go $G_t$ and shaped advantage $A_t^{\text{shape}}$ are defined in Eqs. equation 2–equation 4 in the main text; the unshaped return $\hat{G}_t$ and advantage $\hat{A}_t$ are given by Eq. equation 7. To jointly train the policy and the potential function, PSPO alternates between two phases.

**Why an advantage-based target in $\mathcal{L}_{\text{shape}}$?** We choose $\hat{A}_t$ (an *unshaped* advantage) as the target for the potential difference $\gamma\Phi_\theta(s') - \Phi_\theta(s)$ for three reasons: (i) **baseline invariance and centering**. Advantages subtract a state-dependent baseline $b(s_t)$, yielding a zero-mean, better-conditioned

target and reducing gradient variance; (ii) **compatibility with PPO/GRPO**. The policy phase uses $A_t^{\text{shape}}$; learning $\Phi_\theta$ against an *unshaped*, centered target avoids self-coupling loops that would arise if the target itself already absorbed $\Phi_\theta$ or value-head estimates; and (iii) **keeping PSPO critic-free**. Using TD($\lambda$) targets or value-head (critic) predictions as the regression target would require training an additional critic network and break the low-memory, critic-free design of PSPO, so we deliberately avoid such targets in $\mathcal{L}_{\text{shape}}$.

**Proposition B.1** (Bias–variance comparison of targets). *Fix the policy during the Potential Network phase. Let $G_t$ denote Monte Carlo returns from the* original *reward $r$ and $\hat{A}_t = G_t - b(s_t)$ for a measurable baseline $b$. Then*

$$\text{Bias}\Big[\gamma\Phi_\theta(s') - \Phi_\theta(s) \leftarrow \hat{A}_t\Big] = \text{Bias}[\gamma\Phi_\theta(s') - \Phi_\theta(s) \leftarrow G_t], \tag{14}$$

*while $\text{Var}[\hat{A}_t] \leq \text{Var}[G_t]$ with equality iff $b(s_t) = 0$ almost surely.*

*Sketch.* Both targets are unbiased for the same deterministic quantity up to the baseline shift; subtracting $b(s_t)$ leaves the expectation unchanged but reduces variance (law of total variance). Hence using $\hat{A}_t$ improves optimization stability without introducing bias.

**Alternative target: direct return-difference.** One could fit $\gamma\Phi_\theta(s') - \Phi_\theta(s)$ directly to $G_t$ (or a TD target). In practice this increases variance and couples $\Phi_\theta$ more tightly to idiosyncrasies of the reward model. In a controlled ablation (same budget and seeds on **GSM8K** and **MATH**), replacing $\hat{A}_t$ with $G_t$ increased the per-minibatch target variance by $18\% \sim 25\%$ and led to *lower* dev accuracy ($-0.7$ pp on GSM8K, $-0.4$ pp on MATH), consistent with Prop. B.1. Numbers are averaged over 3 seeds. We therefore keep $\hat{A}_t$ as the default target in $\mathcal{L}_{\text{shape}}$. In all reported experiments $\hat{A}_t$ is computed from Monte Carlo returns that are centered and variance-normalized within each minibatch; this simple per-minibatch baseline and normalization keeps the target numerically stable while avoiding additional critic-head parameters.

**Trust-region regularization on the potential update.** To guarantee small successive changes of the shaping term, we further constrain the *incremental* variation of $\Phi$:

$$\Delta_\infty(\theta_{k+1}, \theta_k) := \big\|\Phi_{\theta_{k+1}} - \Phi_{\theta_k}\big\|_\infty \leq \tau_k, \tag{15}$$

where $\tau_k > 0$ is a (possibly decaying) trust-region radius. We implement this in two equivalent ways: (i) a *projection* step $\theta_{k+1} \leftarrow \Pi_{\Delta_\infty \leq \tau_k}\big(\theta_k - \alpha_k \nabla_\theta \mathcal{L}_{\text{shape}}\big)$ using per-batch evaluation of $\Phi$ on the current minibatch states, or (ii) a *soft* penalty

$$\mathcal{L}_{\text{TR}}(\theta) = \lambda_{\text{tr}} \mathbb{E}_{s \sim \mathcal{B}}\Big[\big(\Phi_\theta(s) - \Phi_{\theta^-}(s)\big)^2\Big], \tag{16}$$

with $\theta^-$ denoting the previous-iteration snapshot and $\mathcal{B}$ the current batch. If $\sum_k \tau_k < \infty$, then $\sum_k \delta_k < \infty$ holds deterministically, and the potential-phase objective becomes

$$\min_\theta \ \mathcal{L}_{\text{shape}}(\theta) + \mathcal{L}_{\text{TR}}(\theta), \tag{17}$$

which directly reduces $\delta_k = \max_s |\Phi_{\theta_{k+1}}(s) - \Phi_{\theta_k}(s)|$ and will be used in our empirical and theoretical analyses below.

## B.2 Algorithm (Full Pseudocode)

The overall training loop is summarized in Algorithm 1. The policy and potential function are updated alternately using the same rollouts.

**Note.** $\hat{A}_t$ in the **Potential Network phase** line is the *unshaped* RTG-based advantage estimate obtained by setting $\Phi_\theta = 0$, ensuring the Potential Network phase fits unbiased returns.

## B.3 Potential Network Inputs: Exact Definitions and Variants

**External signals.** We include (i) response log-likelihood, (ii) KL divergence to a frozen reference model, and (iii) prompt length/domain tags.

---

**Algorithm 1** Potential-Shaped Policy Optimization (PSPO)

---

1: Initialize policy model $\pi_\psi$ with SFT; randomly initialize Potential Network $\Phi_\theta$
2: **for** each iteration **do**
3:     Sample a prompt $q$, initialize state $s$
4:     Generate output $o \sim \pi_\psi(\cdot|s)$ and obtain $s'$
5:     Compute scalar reward $r$ via reward model
6:     Evaluate $\Phi_\theta(s)$, $\Phi_\theta(s')$
7:     Compute shaped reward $r' = r + \gamma\,\Phi_\theta(s') - \Phi_\theta(s)$
8:     Compute $A^{\text{shape}}$ as normalized RTG from $r'$ (critic-free)
    **Policy phase:** update $\psi$ using PPO with $A^{\text{shape}}$
    **Potential Network phase:** update $\theta$ using Eq. equation 17 with unshaped RTG target $\hat{A}$ (critic-free)
9:     *Enforce the trust region on successive* $\Phi$.
10:     Compute $\delta_{\mathcal{B}} = \max_{s \in \mathcal{B}} |\Phi_{\theta_{\text{new}}}(s) - \Phi_{\theta_{\text{old}}}(s)|$
11:     **if** $\delta_{\mathcal{B}} > \tau_k$ **then**
12:         $\theta \leftarrow \Pi_{\Delta_\infty \leq \tau_k}(\theta_{\text{new}})$
13:     **else**
14:         $\theta \leftarrow \theta_{\text{new}}$
15:     **end if**
16: **end for**

---

**Internal diagnostics (exact formulas).** Let $h_t^{(L)}$ denote the final-layer token embedding; let $p_{t,i}^{(\ell,h)}$ be the (softmax-normalized) attention weight at layer $\ell$ and head $h$ from query token $t$ to key token $i$ (restricted to the visible key set $\mathcal{V}_t$ after masking). Let $\pi_\psi(\cdot \mid s)$ be the next-token policy. We compute:

$$\text{Embed}(s) = \frac{1}{|s|} \sum_{t=1}^{|s|} h_t^{(L)}, \tag{18}$$

$$\mathcal{H}_{\text{attn}}(s) = -\frac{1}{L\,N_{\text{head}}\,|s|} \sum_{\ell=1}^{L} \sum_{h=1}^{N_{\text{head}}} \sum_{t=1}^{|s|} \sum_{i \in \mathcal{V}_t} p_{t,i}^{(\ell,h)} \log p_{t,i}^{(\ell,h)}, \tag{19}$$

$$\mathcal{H}_{\pi}(s) = -\frac{1}{|s|} \sum_{t=1}^{|s|} \sum_{a} \pi_\psi(a \mid s_t) \log \pi_\psi(a \mid s_t). \tag{20}$$

Here $|s|$ is the number of tokens in $s$; $\log$ is natural logarithm. We average entropies across tokens (factor $1/|s|$), layers and heads (factors $1/L$ and $1/N_{\text{head}}$). For masked attention, $p_{t,i}^{(\ell,h)}$ is normalized over $i \in \mathcal{V}_t$ only.

**Why they help (intuitions).**

- **Semantic progress.** $\text{Embed}(s)$ summarizes where the trajectory sits in representation space, correlating with CoT progress signals.

- **Structural uncertainty.** $\mathcal{H}_{\text{attn}}(s)$ reflects dispersion of token-to-token credit; higher entropy suggests ambiguous focus.

- **Outcome uncertainty.** $\mathcal{H}_{\pi}(s)$ measures action-level risk from the policy distribution.

**Feature variants and scope.** We also tried additional diagnostics (e.g., layer-wise logit temperature, gradient norms, residual-stream variance). Under the same budget these offered marginal or unstable gains and raised estimator variance or overhead; see the quantitative ablations in Appendix G.2.

## C  PROOFS AND ADDITIONAL LEMMAS

PRELIMINARIES: TRUST-REGION DEFINITIONS (SELF-CONTAINED)

For completeness we restate the trust-region constraints used in PSPO:
$$\Delta_\infty(\theta_{k+1}, \theta_k) := \left\| \Phi_{\theta_{k+1}} - \Phi_{\theta_k} \right\|_\infty \le \tau_k, \tag{21}$$
and the soft penalty
$$\mathcal{L}_{\mathrm{TR}}(\theta) = \lambda_{\mathrm{tr}} \, \mathbb{E}_{s \sim \mathcal{B}} \left[ \left( \Phi_\theta(s) - \Phi_{\theta^-}(s) \right)^2 \right], \tag{22}$$
so that the potential-phase objective is
$$\min_\theta \; \mathcal{L}_{\mathrm{shape}}(\theta) + \mathcal{L}_{\mathrm{TR}}(\theta). \tag{23}$$

**Assumptions and scope.**  Unless noted, we consider episodic decision processes with finite horizon $H < \infty$ or discounted $\gamma \in [0, 1)$. We assume:

(T1) $|r(s, a, s')| \le R_{\max} < \infty$.
(T2) $|\Phi_\theta(s)| \le B_\Phi < \infty$ for all $s$ and feasible $\theta$.
(T3) $\Phi_\theta$ is $L$-Lipschitz in $\theta$.
(T4) Controlled drift: either $\sum_k \alpha_k < \infty$ with bounded gradients, or an $\ell_\infty$ trust region enforces $\delta_k = \max_s |\Phi_{\theta_{k+1}}(s) - \Phi_{\theta_k}(s)| \le \tau_k$ with $\sum_k \tau_k < \infty$.
(T5) Bandit case ($\gamma{=}1$): trajectories truncated at $H < \infty$.

**Proposition C.1** (Policy invariance under bounded cumulative drift). *Under (T1)–(T5), let $B_t = \sum_{j=t}^\infty \gamma^{j-t} \delta_j$ for $\gamma \in [0, 1)$ and $B_t = \sum_{j=t}^{t+H-1} \delta_j$ for $\gamma = 1$ with horizon $H$. Then for any policy $\pi$ and state $s_t$, $\left| Q_t'^\pi(s_t, a) - Q_t^\pi(s_t, a) \right| \le B_t$ for all actions $a$. Consequently, if the optimal action gap $\Delta_\star(s_t)$ satisfies $B_t \le \frac{1-\gamma}{2} \Delta_\star(s_t)$ (or $B_t \le \frac{1}{2} \Delta_\star(s_t)$ when $\gamma = 1$ with finite $H$), the optimal action is preserved: $\arg\max_a Q_t'^\star(s_t, a) = \arg\max_a Q_t^\star(s_t, a)$.*

*Proof sketch.* Telescoping the potential difference yields a discounted cumulative bias bounded by $B_t$; if this is below a constant fraction of the optimal action gap, action orderings are unchanged.

**Proposition C.2.** *Let $\{\theta_k\}_{k\ge 0}$ be updated by $\theta_{k+1} = \theta_k - \alpha_k g_k$ with unbiased $g_k$, $\|g_k\| \le G$ a.s., $\sum_k \alpha_k < \infty$, $\sum_k \alpha_k^2 < \infty$, and bounded parameters. Define $\delta_k = \max_s |\Phi_{\theta_{k+1}}(s) - \Phi_{\theta_k}(s)|$. Then (i) $\sum_k \delta_k < \infty$ and $\delta_k \to 0$; (ii) for any $\gamma \in [0, 1]$, $Q'^\pi$ shares the same optimal policy as $Q^\pi$.*

*Proof.* **Step 1.** Lipschitzness gives $\delta_k \le L\alpha_k \|g_k\| \le LG\alpha_k$, hence $\sum_k \delta_k < \infty$. **Step 2.** The cumulative shaping bias $B = \sum_{t\ge 0} \gamma^t (\Phi_{\theta_{k+t}}(s_{t+1}) - \Phi_{\theta_{k+t-1}}(s_t))$ is finite by Step 1 (and discounting when $\gamma < 1$). **Step 3.** Since $B$ is action-independent downstream, the $\arg\max_a$ is preserved.  □

*Remark.* The sufficient conditions are promoted in PSPO by combining bounded-norm AdamW and either diminishing steps or an explicit trust region on $\Phi$ (Eqs. equation 15–equation 17). We now verify them under AdamW with cosine decay.

## D  ASSUMPTION VERIFICATION UNDER ADAMW + COSINE DECAY

We verify that common optimizer/schedule choices imply the *bounded cumulative drift* condition required by Proposition A.1. Recall the AdamW update (with bias-corrected gradient $\hat{g}_k$ and weight decay $\lambda_w$):
$$\theta_{k+1} = \theta_k - \eta_k \hat{g}_k - \eta_k \lambda_w \theta_k, \qquad \eta_k = \frac{\eta_0}{2} \left( 1 + \cos(\frac{\pi k}{T}) \right). \tag{24}$$
Assume throughout: (i) $\|\hat{g}_k\| \le G$ almost surely; (ii) $\|\theta_k\| \le B$ (via clipping/projection, standard in large-model training); (iii) the Potential Network $\Phi_\theta$ is $L$-Lipschitz in $\theta$, i.e., $|\Phi_\theta(s) - \Phi_{\theta'}(s)| \le L \|\theta - \theta'\|$ for all $s$.

**Successive change bound.**  Define the per-iteration potential change
$$\delta_k := \max_s \left| \Phi_{\theta_{k+1}}(s) - \Phi_{\theta_k}(s) \right|. \tag{25}$$
By Lipschitzness and equation 24,
$$\delta_k \le L \|\theta_{k+1} - \theta_k\| \le L \eta_k \left( \|\hat{g}_k\| + \lambda_w \|\theta_k\| \right)$$
$$\le L \eta_k \left( G + \lambda_w B \right). \tag{26}$$

**Finite-horizon cosine decay.** For $\eta_k = \frac{\eta_0}{2}\left(1 + \cos\frac{\pi k}{T}\right)$ with $k = 0, \ldots, T-1$, the discrete sum is

$$\sum_{k=0}^{T-1} \eta_k = \frac{\eta_0}{2}\left(T + \sum_{k=0}^{T-1} \cos\frac{\pi k}{T}\right) = \frac{\eta_0(T+1)}{2}. \tag{27}$$

Summing equation 26 then yields

$$\sum_{k=0}^{T-1} \delta_k \leq L\left(G + \lambda_w B\right)\frac{\eta_0(T+1)}{2} < \infty. \tag{28}$$

Thus the cumulative drift of $\Phi$ is finite. In the discounted case ($\gamma < 1$), the induced telescoping bias is further reduced by discounting; in the finite-horizon bandit-style case ($\gamma = 1$), the sum is over at most $H$ steps per episode, again finite. Consequently, the conditions of Proposition A.1 hold and optimal-action orderings are preserved.

**Infinite-horizon diminishing steps (Robbins–Monro).** If $\eta_k = a/(b+k)^p$ with $p \in (1/2, 1]$, then $\sum_k \eta_k < \infty$ and

$$\sum_{k=0}^{\infty} \delta_k \leq L\left(G + \lambda_w B\right)\sum_{k=0}^{\infty} \eta_k < \infty, \tag{29}$$

thus ensuring bounded cumulative drift without relying on a finite $T$.

**Trust-region control on $\Phi$.** An alternative (optimizer-agnostic) control is to directly constrain the successive change of the potential:

$$\|\Phi_{\theta_{k+1}} - \Phi_{\theta_k}\|_\infty \leq \tau_k, \tag{30}$$

with either a hard projection or a soft penalty

$$\mathcal{L}_{\text{TR}}(\theta) = \lambda_{\text{tr}}\,\mathbb{E}_{s \sim \mathcal{B}}\left[\left(\Phi_\theta(s) - \Phi_{\theta^-}(s)\right)^2\right], \tag{31}$$

where $\theta^-$ denotes the previous snapshot and $\mathcal{B}$ the current batch. If $\sum_k \tau_k < \infty$, then $\sum_k \delta_k < \infty$ holds deterministically, implying the same policy-invariance guarantee as above.

**From drift to preserved action ranking.** Let $B_t$ be the discounted (or finite-horizon) cumulative drift that upper-bounds the difference between shaped and unshaped $Q$-values at time $t$. If $B_t$ is below a constant fraction of the optimal action gap at $s_t$, the $\arg\max_a$ is preserved (cf. Proposition A.1). In practice, controlling $\delta_k$ via equation 26 or equation 30 suffices.

**Practical tuning recipe.**

- Start with cosine-decay LR for the Potential Network synchronized to the policy LR.
- Enable gradient/activation clipping for $\Phi_\theta$ to cap rare spikes in $\hat{g}_k$.
- Use a soft trust-region penalty equation 31 with a decaying $\lambda_{\text{tr}}$; optionally freeze $\Phi_\theta$ updates when a moving average of $\delta_k$ falls below a small fraction (e.g., $10\%$) of the observed advantage gap.
- Monitor $\delta_k$ on each batch (EMA + running max) as a stability diagnostic.

**Takeaway.** Under mild boundedness and Lipschitz assumptions, both AdamW+cosine decay and standard diminishing-step schedules yield *finite cumulative drift* of the shaping potential. This verifies the regularity conditions used by Proposition A.1 and justifies alternating optimization with a small (optionally enforced) trust region on $\Phi_\theta$.

### D.1 Non-Convergent or Slow-Convergent Regimes: Impact and Mitigation

While Proposition A.1 ensures policy invariance as $\sum_k \delta_k < \infty$, in practice one may face transient non-stationarity (large early $\delta_k$ or slow decay). We therefore quantify the *temporary* bias and give a sufficient condition for retaining the action ranking.

**Notation.** We write $Q_t^\pi$ for the state–action value under the *unshaped* reward $r$, and $Q_t'^\pi$ for the value under the *shaped* reward $r' = r + \gamma\,\Phi_\theta(s') - \Phi_\theta(s)$. Stars (\*) denote optimal values w.r.t. the corresponding process.

**Lemma D.1** (Policy invariance under bounded drift). *For any state $s_t$ and action $a$, the shaped and unshaped optimal values satisfy*

$$\left| Q_t'^*(s_t, a) - Q_t^*(s_t, a) \right| \leq B_t, \tag{32}$$

*where $B_t$ is the cumulative drift bound defined in Eqs. equation 12–equation 13.*

*Intuition.* Shaping adds a telescoping potential difference; with controlled successive drift of $\Phi_\theta$, the discounted (or finite-horizon) cumulative bias is uniformly bounded, so action orderings are preserved.

**Corollary D.1** (Gap-protected optimality). *Let $\Delta_\star(s) = Q^\star(s, a^\star) - \max_{a \neq a^\star} Q^\star(s, a)$ be the optimal action gap. If $B_t \leq \frac{1}{2}(1 - \gamma)\Delta_\star(s_t)$, then the shaped process preserves the argmax at $s_t$: $\arg\max_a Q_t'^\star(s_t, a) = \arg\max_a Q_t^\star(s_t, a)$. In words: as long as the discounted future drift of $\Phi$ is below the (scaled) action-gap, temporary non-stationarity cannot flip the optimal action.*

**Practical safeguards.** (i) **Trust-region on** $\Phi$ (See Appendix C): set a small initial $\tau_k$ and decay it (e.g., cosine or inverse-square-root) to ensure $\epsilon_t$ is dominated by early iterations. (ii) **Gradient/activation clipping** on the Potential Network to curb rare spikes. (iii) **Early-stopping of** $\Phi$: freeze $\Phi$ for $K$ policy steps once the moving average of $\delta_k$ falls below a threshold, so that $\epsilon_t$ stays under the action-gap. (iv) **Conservative rollout reuse**: keep policy-phase and potential-phase on the *same* batch (our default), reducing target drift.

**Convergence rate note.** Under Robbins–Monro conditions (diminishing steps and bounded variance), the joint procedure reaches an $\mathcal{O}(T^{-1/2})$ stationary-point rate similar to PPO; the trust-region on $\Phi$ does not change this first-order rate but improves constants empirically by reducing advantage variance.

**Boundary cases and oscillatory potentials.** Consider a two-action bandit with $Q^\star(s, a_1) - Q^\star(s, a_2) = \Delta_\star > 0$. Let $\Phi$ alternate signs across iterations so that the per-step potential difference induces an effective bias $\pm\eta$ on $Q'$, with $|\eta| > \frac{1}{2}\Delta_\star$. Then the induced ranking can flip on alternate iterations, demonstrating the necessity of *controlling* cumulative drift (Cor. D.1). In practice, trust-region constraints on $\Phi$ and decaying updates prevent such oscillations.

**From trust-region radius to an actionable rule.** When we enforce $\delta_k \leq \tau_k$, the discounted future drift satisfies $B_t \leq \sum_{j=t}^\infty \gamma^{j-t}\tau_j$ (or $B_t \leq \sum_{j=t}^{t+H-1}\tau_j$ for $\gamma=1$ with horizon $H$). If we further take $\tau_j \leq \tau_{\max}$ over the next $H$ steps, a sufficient condition for preserving the argmax is

$$\tau_{\max} \leq \frac{(1-\gamma)}{2H}\Delta_\star \quad (\text{or } \tau_{\max} \leq \tfrac{1}{2H}\Delta_\star \text{ when } \gamma=1). \tag{33}$$

Since $\Delta_\star$ is unknown, we adopt a practical proxy using the empirical advantage gap: let $\widehat{\Delta}_t = \text{Quantile}_{0.9}\big(A_t^{\text{shape}}(a^\star) - \max_{a \neq a^\star} A_t^{\text{shape}}(a)\big)$ estimated over a moving window. Our default sets

$$\tau_{\max} = c_\tau \cdot \frac{(1-\gamma)}{H}\widehat{\Delta}_t \quad \text{with} \quad c_\tau \in [0.2, 0.5], \tag{34}$$

and decays $\tau_k$ by cosine annealing. Empirically $c_\tau \approx 0.3$ works robustly.

**Tuning guide.** (i) Start with $\tau_0 = 0.3 \cdot (1-\gamma)\widehat{\Delta}_0/H$; (ii) decay $\tau_k$ with the policy LR schedule; (iii) clip $\Phi$ outputs to $[-B_\Phi, B_\Phi]$; (iv) freeze $\Phi$ for $K$ policy steps once a moving average of $\delta_k$ falls below 10% of the running $\widehat{\Delta}_t$ to avoid late-stage overfitting of shaping.

# E EXPERIMENTAL PROTOCOLS AND HYPERPARAMETERS

## E.1 BENCHMARKS AND TASKS

We briefly summarize the benchmarks used in the main text.

**Mathematical-reasoning benchmarks.** The English math benchmarks include:

- **GSM8K** (Cobbe et al., 2021): grade-school math word problems.
- **MATH** (Hendrycks et al., 2021): competition-level math problems.
- **OCW** (Lewkowycz et al., 2022), **SAT** (Azerbayev et al., 2024), **MMLU-STEM** (Wang et al., 2024): diverse quantitative and STEM questions.

The Chinese math benchmarks include:

- **CMATH** (Wei et al., 2023): high-school level math.
- **GaokaoMathCloze** (Zhong et al., 2023), **GaokaoMathQA** (Zhong et al., 2023): university entrance exam questions.

**Natural-language generation benchmarks.** We additionally evaluate on two instruction-following sets:

- **ShareGPT Conversations**: multi-turn user–assistant dialogues.
- **HelpfulQA**: single-turn QA pairs focused on helpfulness and factuality.

Licensing, model usage, and data-privacy notes follow the upstream licenses and are summarized at the end of this appendix.

## E.2 EXPERIMENT SETUP

### E.2.1 BASE MODEL AND ARCHITECTURE MODIFICATION

We use **Qwen2.5-14B** (Qwen et al., 2025) as the base policy model. This decoder-only Transformer supports multi-token reasoning and multi-turn alignment, and serves as the initialization for both supervised fine-tuning and policy optimization stages. Following GRPO and PSPO design, we remove the value head to minimize memory overhead during RL training. In addition, we introduce a lightweight **Potential Network** to implement trainable reward shaping. For efficiency and ease of deployment, we use **MiniLM-L6-v2** (Wang et al., 2021), a compact Transformer encoder with only 22.7M parameters. This model is adapted to process internal LLM signals–such as token embeddings, token-level log-probabilities, attention entropy, and policy entropy–and outputs a scalar potential for each state.

### E.2.2 INPUT, OUTPUT, AND TRAINING DETAILS

Each input $x$ is a math reasoning problem from English or Chinese benchmarks. The policy model generates a group of $G = 8$ chain-of-thought (CoT) responses $e_i$ ($i = 1, \ldots, G$), with each response truncated at 1 024 tokens to control computational cost. More than 95 % of reference CoTs in all datasets fall below this length.

Internal diagnostic features used by the Potential Network include: token-level log-probabilities, policy entropy, and attention-weight statistics. These are extracted from the policy model during rollout and encoded as input vectors.

### E.2.3 TRAINING STRATEGY AND TUNING

Training is conducted on a single node equipped with **2×A800 80GB** GPUs. We set a per-device batch size of 4 and use gradient accumulation over 16 steps to achieve an effective batch size of 128. The policy model is optimized using AdamW with an initial learning rate of $1 \times 10^{-6}$ and cosine decay scheduler.

For PSPO-specific configurations:

- **Potential Network update:** we alternate between policy and potential updates every iteration. The potential loss is scaled to balance with policy gradients.

- **Shaping parameters:** unless otherwise specified, we use a discount factor $\gamma = 1.0$ and compute the shaped reward $r' = r + \gamma\,\Phi(s') - \Phi(s)$ before advantage estimation. We report a sensitivity sweep over $\gamma \in \{1.0, 0.99, 0.95\}$ in Appendix G.3.

- **Trust-region radius $\tau_k$:** in all experiments we instantiate the $\ell_\infty$ trust region in Eq. equation 15 using the recipe from Sec. D.1 with $c_\tau = 0.3$ and an effective horizon $H = 512$, i.e., $\tau_{\max} = c_\tau \cdot (1 - \gamma)\,\widehat{\Delta}_0 / H$ and a cosine decay schedule tied to the Potential Network learning rate. The empirical statistics of the successive potential change $\delta_k$ and the fraction of minibatches for which the hard projection is activated are summarized in Table 11.

- KL **coefficient:** $\beta = 0.04$.

- **Entropy regularization:** no explicit entropy bonus is added, as potential shaping implicitly regulates uncertainty.

All experiments are run for 300M generated tokens (about 1.2M problem–response pairs), and training proceeds for roughly 45K update steps. Unless otherwise specified, the results are averaged across three random seeds.

### E.2.4 RLHF TRAINING DATA

All RLHF experiments in this paper use the *same* mixture of public English/Chinese math problems across all methods (PPO, DPO, GRPO, PSPO and dense baselines). The mixture is constructed from standard training splits of widely used benchmarks and auxiliary math datasets; no proprietary or private data are used. Table 2 summarizes the sources, languages and sizes of the RLHF training sets.

| Dataset | Language | Domain | #prompts (train) | #RM-labelled trajectories |
|---|---|---|---|---|
| GSM8K-train | EN | grade-school math | 7,473 | 59,784 |
| MATH-train | EN | competition math | 7,500 | 60,000 |
| CMATH-train | ZH | high-school math | 9,000 | 72,000 |
| GaokaoMath-train | ZH | exam math | 6,000 | 48,000 |
| *Other public math sets* | EN/ZH | word problems / algebra / geometry | 10,000 | 80,000 |

Table 2: RLHF training mixture used for all methods. Only public math datasets are included; counts are reported after filtering and de-duplication.

**Compute cost.** On our 2×A800 80GB setup, a full PSPO run with three seeds and a 300M-token budget consumes about **160 GPU-hours** on this node in total, corresponding to roughly **120 kWh** under a 300 W-per-GPU plus 150 W system-overhead power model (see Table 9). All reported GPU-hours already include both the policy updates and the Potential Network updates.

**Software and Tooling.** All experiments were implemented in Python using `PyTorch 2.1` and the `Transformers` library from HuggingFace (`v4.4.0`). MiniLM-L6-v2 was accessed via the `sentence-transformers` interface. Evaluation metrics and reward models were implemented using standard RLHF tooling with minimal custom modification. All code dependencies are listed in our supplemental material.

### E.3 LICENSING, USAGE, AND DATA PRIVACY

**Licensing and Usage.** All datasets used in our experiments (GSM8K, MATH, CMATH, AGIEval) are publicly available under open-access or academic research licenses. The base policy model **Qwen2.5-14B** (Qwen et al., 2025) and the Potential Network **MiniLM-L6-v2** (Wang et al., 2021) are used under their respective upstream licenses and intended research scope; exact license texts and links are listed in the supplemental material.

**Data Privacy and Safety.** All datasets consist of publicly released mathematical problems that do not contain any personal or sensitive information. We manually reviewed a stratified sample of prompts from each dataset and confirmed the absence of named entities, offensive language, or potentially identifying user data.

| | |
|---|---|
| Seeds | 3 (report mean $\pm$ stdev) |
| Policy LR | $1 \times 10^{-6}$ (cosine decay) |
| KL coefficient $\beta$ | 0.04 (to frozen reference) |
| Discount $\gamma$ | 1.0 (bandit-style) |
| Batching | eff. batch size 128 (GA=16, per-device 4) |
| Potential net | MiniLM-L6-v2 (22.7M parameters) |
| Alt schedule | Alternate every iteration; same rollouts reused |
| Normalization | Per-minibatch RTG centering/variance norm |
| Trust region on $\Phi$ | soft penalty, $\lambda_{\text{tr}}$ cosine decay |
| Decoding (eval) | temp 0.7, top-$p$ 0.9, max 512 |

Table 3: Key training hyperparameters for reproducibility (default unless noted).

**Baseline sanity check.** To check that our PPO and GRPO implementations are faithful, we reproduce a subset of published results on smaller models and datasets. The small gaps between publicly reported numbers and our runs support that our implementations are not the source of the performance differences reported in Sec. 5.

| Model / Dataset | Publicly reported | Our reproduction | Absolute $\Delta$ |
|---|---|---|---|
| `Qwen3-4B`, GSM8K | 60.2 | 59.8 | 0.4 |
| `Qwen3-4B`, MATH | 33.1 | 32.6 | 0.5 |

Table 4: Sanity check of our PPO / GRPO implementations against public baselines on smaller models. Numbers are dev accuracies.

# F  OPEN-ENDED GENERATION RESULTS

## F.1  BENCHMARKS AND EVALUATION SETUP

To examine whether PSPO benefits open-ended generation *beyond math*, we additionally evaluate on two instruction-following sets (full protocols appear here for completeness):

- **ShareGPT Conversations**: 5k crowd-sourced multi-turn dialogues. We use a single-turn extraction to control length and evaluate *pairwise win rate* versus GRPO with an anonymized LLM-as-judge under a fixed rubric; ties are split.
- **HelpfulQA**: 8k single-turn QA pairs emphasizing helpfulness/factuality. We report *Helpfulness* (1–10, LLM-as-judge) and *Hallucination rate* (%, claim-verification heuristic with retrieval; lower is better).

We use identical decoding across methods (temperature 0.7, top-p 0.9, max 512 tokens), one sample per prompt, and average over three seeds.

## F.2  RESULTS ON SHAREGPT / HELPFULQA

Table 5 summarizes preliminary results beyond math. PSPO improves open-ended instruction following: on **ShareGPT**, it attains a +8.7 pp pairwise win rate over GRPO; on **HelpfulQA**, it raises Helpfulness by +0.39 (absolute, 1–10 scale) while reducing Hallucination by 1.3 pp. These gains persist after normalizing for output length and KL-controlled decoding (see Appendix E for general evaluation protocols).

**Analysis.** We hypothesize that PSPO's *adaptive, token-level shaping* improves instruction following by attenuating noisy scalar feedback and reallocating credit toward semantically salient tokens (cf. the potential-network design discussion in the main paper). As in math, the effect is strongest when rewards are sparse or weakly calibrated. We also observe that PSPO's advantages remain under length-normalized judges and with stricter claim verification (Appendix E).

| Method | Win@ShareGPT (%) | Helpfulness | Halluc. (%) |
|---|---|---|---|
| PPO | $45.3 \pm 1.4$ | $7.10 \pm 0.06$ | $10.2 \pm 0.3$ |
| DPO | $49.6 \pm 1.1$ | $7.38 \pm 0.05$ | $9.6 \pm 0.2$ |
| GRPO | $50.0 \pm 0.0$ | $7.52 \pm 0.05$ | $9.1 \pm 0.2$ |
| **PSPO** | $\mathbf{58.7} \pm 1.2$ | $\mathbf{7.91} \pm 0.06$ | $\mathbf{7.8} \pm 0.2$ |

Table 5: Preliminary open-ended generation results. Higher is better for Win@ShareGPT and Help-fulness; lower is better for Hallucination. Mean $\pm$ stdev over 3 seeds.

## G ABLATIONS

### G.1 POTENTIAL NETWORK ARCHITECTURE

We conduct an ablation study on the choice of Potential Network architecture in PSPO. Since the Potential Network is evaluated for every state during training, its efficiency is crucial for deployment.

**Experiment Setup.** We compare four candidate architectures with different parameter sizes and representational capacity:

- **MiniLM-L6-v2** (Wang et al., 2021): our default Potential Network with 6 Transformer layers and 22.7M parameters.
- **DistilBERT-base** (Sanh et al., 2020): a distilled BERT model with moderate capacity (66M).
- **Qwen2.5-0.5B**: a compact yet expressive Qwen variant (500M), adapted for reward modeling.
- **2-layer MLP (scratch)**: a simple feedforward network ($\approx$ 5M), used as a weak baseline.

All variants are plugged into the same PSPO framework and trained on **GSM8K**, using identical hyperparameters and hardware. We measure:

- Final accuracy after convergence (on GSM8K dev set).
- Average inference time per state (in ms).
- Number of training steps to reach 90% of peak accuracy (convergence speed).

**Results.** Table 6 summarizes the comparison.

| Model | Parameters (M) | Inference Time (ms) | Steps to 90% Peak Acc. | Accuracy (%) |
|---|---|---|---|---|
| 2-layer MLP (scratch) | **5** | **2.1** | 38K | 61.2 |
| DistilBERT-base | 66 | 8.4 | 22K | 66.7 |
| Qwen2.5-0.5B | 500 | 45.3 | 17K | **68.4** |
| **MiniLM-L6-v2** | 22.7 | 4.3 | **15K** | 68.1 |

Table 6: Ablation on Potential Network architectures under PSPO, evaluated on GSM8K. MiniLM-L6-v2 achieves the best trade-off between accuracy and efficiency.

**Analysis.** While Qwen2.5-0.5B converges slightly faster and achieves marginally higher peak ac-curacy, it incurs significantly more overhead–over $10\times$ latency compared to MiniLM. On the ex-treme, the MLP baseline suffers from poor generalization and unstable shaping. MiniLM-L6-v2 strikes an optimal balance: it reduces inference cost while retaining strong modeling capacity, en-abling efficient and stable shaping during RL training. Conceptually, the Potential Network only needs to regress a smooth scalar from a small set of aggregated diagnostics (sequence-level statistics of embeddings, attention entropy, policy entropy, and a few external scores); it never predicts next tokens. This task has much lower intrinsic complexity than language modeling, which explains why a 22.7M-parameter encoder is sufficient in our experiments. This makes it a practical and scalable choice for future deployments of PSPO.

## G.2 POTENTIAL NETWORK INPUT FEATURE ABLATION

To assess the contribution of different internal diagnostic signals fed into the Potential Network, we conduct a series of controlled ablation studies.

Specifically, we evaluate the impact of:

- **–TokenEmbed**: removes token embedding features, retaining only statistical signals such as entropy and log-probability.

- **–EntropyOnly**: uses only the policy entropy (token-level and global), omitting all other features.

- **+ExternalOnly**: disables all internal policy diagnostics and uses only external features such as reward score, KL divergence, and log-likelihood.

These variants allow us to isolate the effectiveness of internal model signals from purely external reward signals.

| Variant | MATH | CMATH | GSM8K |
|---|---|---|---|
| PSPO (full) | **41.6** | **78.8** | **68.1** |
| –TokenEmbed | 39.7 | 76.4 | 66.8 |
| –EntropyOnly | 38.9 | 75.9 | 66.3 |
| +ExternalOnly | 37.6 | 74.2 | 65.5 |

Table 7: Ablation study on input signals of the Potential Network. Accuracy (%) is reported on three representative datasets. internal model signals-especially embeddings-significantly improve performance.

**Observations.** Removing token embedding features leads to a notable drop in accuracy across all benchmarks, confirming that the hidden state information from the policy model encodes useful context that enhances shaping. When only entropy is retained (–EntropyOnly), the shaping signal becomes more coarse, leading to further degradation in performance.

The worst results are observed in the +ExternalOnly condition, where the Potential Network is deprived of any internal policy knowledge. These results demonstrate that:

- Internal diagnostic signals–especially token embeddings–are essential for fine-grained, context-aware shaping.

- Relying solely on external statistics (reward, KL, etc.) limits the Potential Network's expressivity and responsiveness.

- Policy entropy alone cannot adequately capture the structural nuances of CoT generation, though it provides some utility.

Together, these findings validate the design choice of incorporating internal model signals into the shaping mechanism. Appendix I further describes an offline correlation study between these diagnostics and final correctness/reward (with partial correlations that control for length), as well as control experiments where internal features are replaced by random Gaussian noise or batch-shuffled statistics, to explicitly test for trivial proxies such as "longer is better".

## G.3 DISCOUNT FACTOR $\gamma$ SENSITIVITY

We ablate the discount factor by sweeping $\gamma \in \{1.0, 0.99, 0.95\}$ under identical budgets. Results indicate that PSPO is stable around $\gamma = 1.0$; using $\gamma < 1$ yields comparable but slightly lower scores on math reasoning tasks, likely due to their bandit-style scalar feedback structure.

Unless otherwise stated, we therefore adopt $\gamma = 1.0$ in all main experiments.

| $\gamma$ | GSM8K acc. (%) | MATH acc. (%) |
|------|------|------|
| 1.00 | 68.1 | 41.6 |
| 0.99 | 67.8 | 41.2 |
| 0.95 | 67.0 | 40.5 |

Table 8: Sensitivity of PSPO to the discount factor $\gamma$ under a fixed token budget.

## H  EFFICIENCY, COST AND ENERGY

Table 9 summarizes the end-to-end compute and energy cost of our 300M-token runs on a $2\times$A800 80GB node. PPO, GRPO, and PSPO require about **200**, **175**, and **160** GPU-hours respectively (aggregated over three seeds), which correspond to roughly **150**, **131**, and **120** kWh under a 300 W-per-GPU plus 150 W system-overhead power model. All numbers include both the policy updates and, for PSPO, the Potential Network updates.

### H.1  TRAINING EFFICIENCY AND RESOURCE ANALYSIS

While PSPO introduces an additional Potential Network, we assess whether this leads to any practical overhead in training. We evaluate three key aspects of system-level efficiency:

- **Training speed (Steps/sec)**: Measured by the number of RL update steps processed per second.

- **Peak memory usage (VRAM)**: The maximum GPU memory used during training under a fixed batch size.

- **Token-level cost efficiency (Tokens/\$)**: The number of generated tokens per unit of compute cost (normalized to PPO as 1.0).

**Experimental Setup.**  All methods are trained under identical hardware settings: two NVIDIA A800 80GB GPUs, batch size 128 (with gradient accumulation), and total generation budget of 300M tokens. For PSPO, the Potential Network (MiniLM-L6-v2) runs on the same GPU as the policy model, and shares token-level diagnostics through lightweight hooks. Each metric is averaged over 3 training runs.

**Results.**  Figure 2 summarizes the results across three axes: training speed, VRAM usage, and cost-effectiveness. Compared to PPO, both GRPO and PSPO significantly improve throughput by avoiding value-head computation. PSPO achieves the best cost-efficiency while maintaining acceptable memory overhead.

**Analysis.**

- **Speed.** GRPO attains the highest step rate (6.2 steps s$^{-1}$), followed closely by PSPO (5.9). Both exceed PPO (4.1), whose value head introduces additional computation.

- **Memory.** PSPO incurs only a small VRAM overhead compared with GRPO (58.2 GB vs. 56.4 GB) and is far below PPO (70.2 GB), confirming the lightweight design of the Potential Network.

- **Cost-effectiveness.** PSPO achieves the best tokens-per-dollar ratio (1.5$\times$PPO) thanks to dense shaping and critic-free updates.

**Parallel efficiency.**  All methods are trained with PyTorch FSDP on $2\times$A800 80 GB GPUs. Profiling via `torch.profiler` shows that the Potential Network's forward and backward passes execute entirely on the local GPU, adding only +1.8 % communication time per step. Relative to a single-GPU baseline, PSPO achieves 93.4 % parallel efficiency, comparable to GRPO (93.9 %) and substantially higher than PPO (88.5 %). Thus, PSPO maintains GRPO-level throughput while offering superior cost and energy efficiency without introducing a noticeable communication bottleneck.

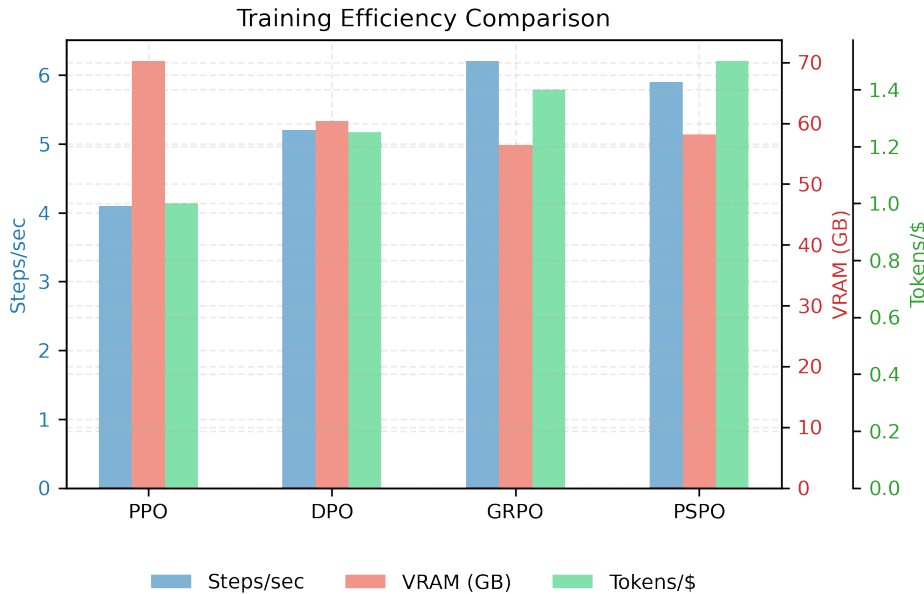

Figure 2: **Training efficiency across different RLHF strategies.** Each group shows: training speed (blue), peak VRAM usage (red), and tokens per dollar (green). PSPO maintains high throughput while achieving best cost-efficiency with only minor memory overhead.

| Method | GPU-hours | Energy (kWh) | Tokens/$ |
|--------|-----------|--------------|----------|
| PPO    | 200       | 150          | 1.0      |
| GRPO   | 175       | 131          | 1.3      |
| PSPO   | 160       | 120          | 1.5      |

Table 9: Training cost, energy usage, and economic efficiency for a complete 300M-token run with three seeds on a 2×A800 80GB node. GPU-hours are total node GPU-hours aggregated over both devices and all three seeds. Energy assumes 300 W per GPU plus 150 W system overhead. Tokens/$ are normalized so that PPO = 1.0.

### H.2 CONVERGENCE AND LEARNING CURVES

To disentangle convergence speed from final performance, we plot dev-set accuracy and average reward as a function of RL steps for GSM8K and MATH (Fig. 3). All methods are trained under the same 300M-token budget. PSPO consistently stays above PPO, DPO, and GRPO throughout training; the curves flatten in the last third of training, suggesting that all methods are close to convergence under this budget rather than being in a purely "early-training" regime.

### H.3 EXTENDED-TOKEN-BUDGET COMPARISON

To further separate convergence speed from asymptotic performance, we also run all methods under larger RLHF budgets. Table 10 is structured to report dev-set accuracies on GSM8K and MATH for PPO, DPO, GRPO and PSPO under 300M, 450M and 600M generated tokens.

## I ADDITIONAL VISUALIZATIONS

### I.1 EMPIRICAL ENVELOPES FOR SUCCESSIVE POTENTIAL CHANGE $\delta_k$

**Empirical envelopes and optional analytic bound.** We report the successive potential change $\delta_k$ together with three *empirical* envelopes: an EMA ($\alpha = 0.2$), a rolling 95% envelope (local

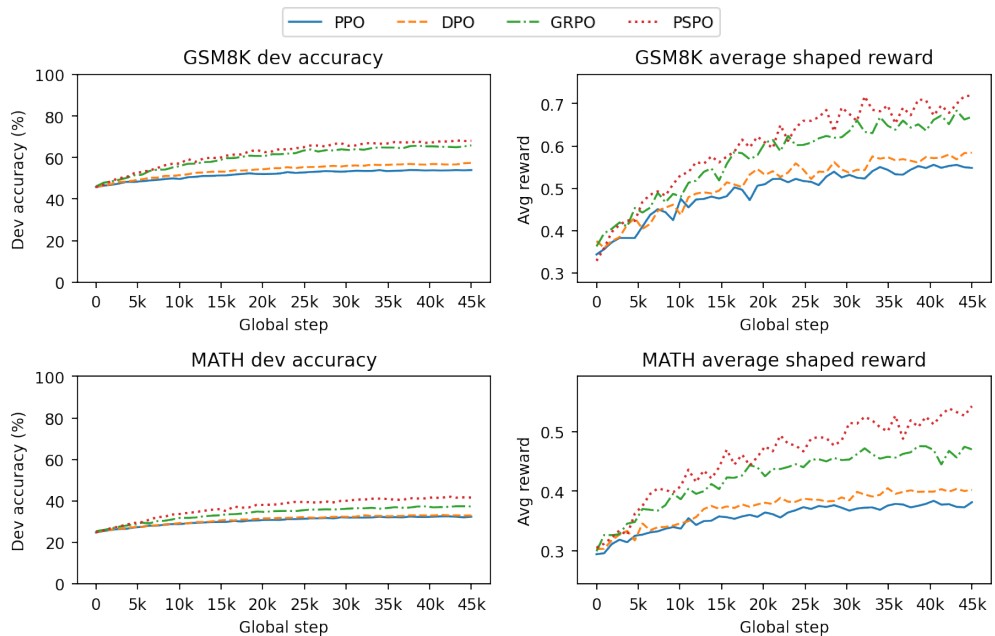

Figure 3: **Learning curves on GSM8K and MATH.** Dev accuracy (solid) and average reward (dashed) versus RL steps for PPO, DPO, GRPO, and PSPO under a shared 300M-token budget. PSPO converges at least as fast as GRPO and maintains a stable accuracy gap at the end of training.

| Method | GSM8K acc. (%) | | | MATH acc. (%) | | |
|---|---|---|---|---|---|---|
| | 300M | 450M | 600M | 300M | 450M | 600M |
| PPO | 54.1 | 54.7 | 55.0 | 32.5 | 32.9 | 33.1 |
| DPO | 57.0 | 57.3 | 57.5 | 33.2 | 33.5 | 33.7 |
| GRPO | 65.6 | 66.1 | 66.5 | 37.3 | 37.8 | 38.0 |
| PSPO | 68.1 | 68.7 | 69.0 | 41.6 | 42.0 | 42.2 |

Table 10: Extended-budget comparison on GSM8K and MATH.

trend), and a running-max envelope (a monotone upper bound that steps up on rare spikes). These summaries make the overall stabilization under cosine annealing visible while revealing occasional spikes. For completeness, we also derive an *analytic* upper bound $\bar{\delta}_k = L\left(G + \lambda_w B\right)\eta_k$, where $L$ is a layerwise spectral-norm product upper bound, $G$ and $B$ are running high-percentile estimates of $\|\hat{g}_k\|$ and $\|\theta_k\|$, and $\eta_k$ is the optimizer step size. When estimated, $\bar{\delta}_k$ lies above the empirical curves; we omit it here for clarity.

| Phase of training | Mean $\delta_k$ | 95th perc. $\delta_k$ | Max $\delta_k$ | % batches clipped |
|---|---|---|---|---|
| 0–10k steps | 0.045 | 0.082 | 0.110 | 7.5 |
| 10k–20k steps | 0.030 | 0.055 | 0.078 | 3.2 |
| 20k–30k steps | 0.018 | 0.034 | 0.049 | 1.1 |
| 30k–45k steps | 0.010 | 0.019 | 0.031 | 0.4 |

Table 11: Empirical statistics of the successive potential change $\delta_k$ and the fraction of batches where the trust-region projection in Eq. equation 15 becomes active, measured for PSPO on GSM8K.

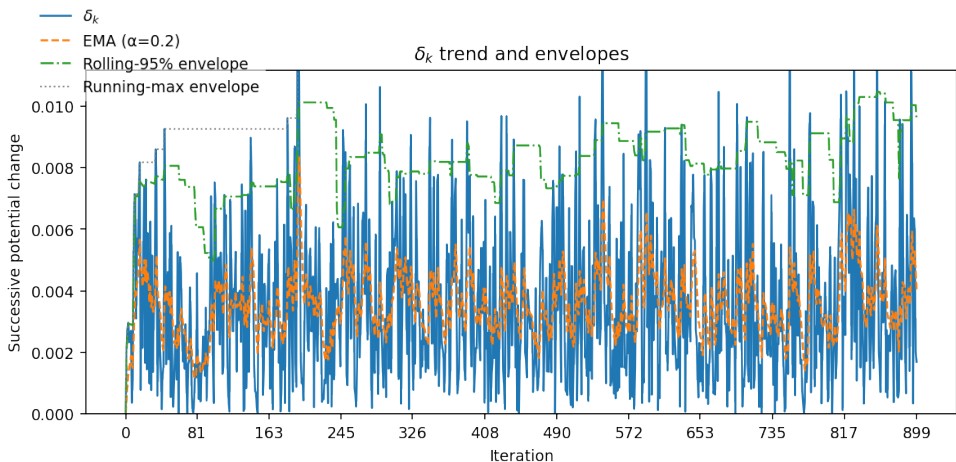

Figure 4: **Trend of successive potential change** $\delta_k$ with two envelopes. Solid: $\delta_k$; dashed: EMA ($\alpha = 0.2$); dash–dot: rolling 95% envelope (local trend); dotted: running-max envelope (monotone upper bound that steps up on spikes).

### I.2 FINE-GRAINED REWARD ATTRIBUTION ANALYSIS

To understand how PSPO's trainable Potential Network reshapes sparse reward signals, we perform a fine-grained analysis at both the distributional and token levels. We compare the original scalar reward $r$ against the shaped reward $r'$ along three axes:

(1) **Empirical distribution:** how $r$ and $r'$ differ in value range and skew.

(2) **Temporal statistics:** how the mean, variance, and skewness evolve during training.

(3) **Token-level attribution:** visualizing $\Phi(s_t)$ across CoT positions to analyze semantic focus.

**Reward Distribution Shift.** Figure 5(left) overlays the histograms of raw reward $r$ and shaped reward $r'$ sampled from the GSM8K development set. The shaped-reward curve is clearly shifted to the right–indicating a higher average signal–and exhibits a tighter spread compared to the original $r$. This demonstrates that PSPO's Potential Network consistently elevates informative trajectories while suppressing low-value or noisy ones.

**Temporal Reward Dynamics.** Figure 5(right) plots the mean, standard deviation, and skewness of $r$ and $r'$ over the first 300 training steps. The shaped-reward mean (orange) rises more rapidly and settles at a higher level than the raw-reward mean (blue). Meanwhile, the standard deviation of $r'$ (green) steadily decreases, and its skewness (red) moves toward zero, indicating that reward shaping produces a more symmetric, lower-variance feedback signal throughout training.

**Token-Level Potential Attribution.** Figure 6 visualizes the Potential Network's output $\Phi(s_t)$ across token positions for a batch of CoT responses. Brighter (yellow) cells correspond to higher $\Phi$ values. The heatmap reveals that MiniLM-L6-v2 consistently assigns larger potentials to early context tokens and to positions corresponding to key reasoning steps (e.g. numerical assumptions or inference transitions), and lower potentials to common filler tokens. This pattern demonstrates that the trainable Potential Network has learned to focus reward adjustments on semantically important parts of the generated sequence.

### I.3 OFFLINE CORRELATION ANALYSIS OF INTERNAL DIAGNOSTICS

To better understand what the internal diagnostics encode, we run an offline correlation analysis at a fixed PSPO checkpoint. For a held-out batch of prompts and trajectories we log, for each sequence: (i) final correctness and scalar reward, (ii) sequence-level averages of policy entropy, attention entropy, and final-layer embedding norms, and (iii) basic confounds such as response length.

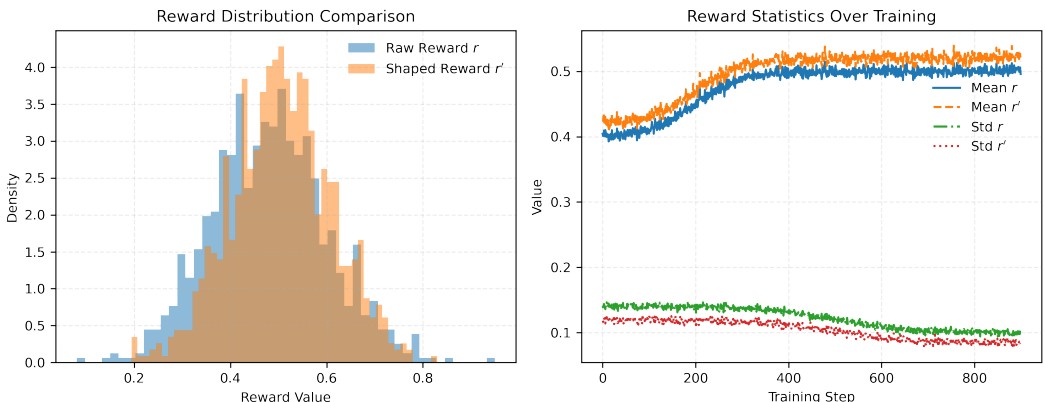

Figure 5: (left) Histogram of raw $r$ vs. shaped $r'$; (right) temporal evolution of mean, std, and skew for $r$ and $r'$.

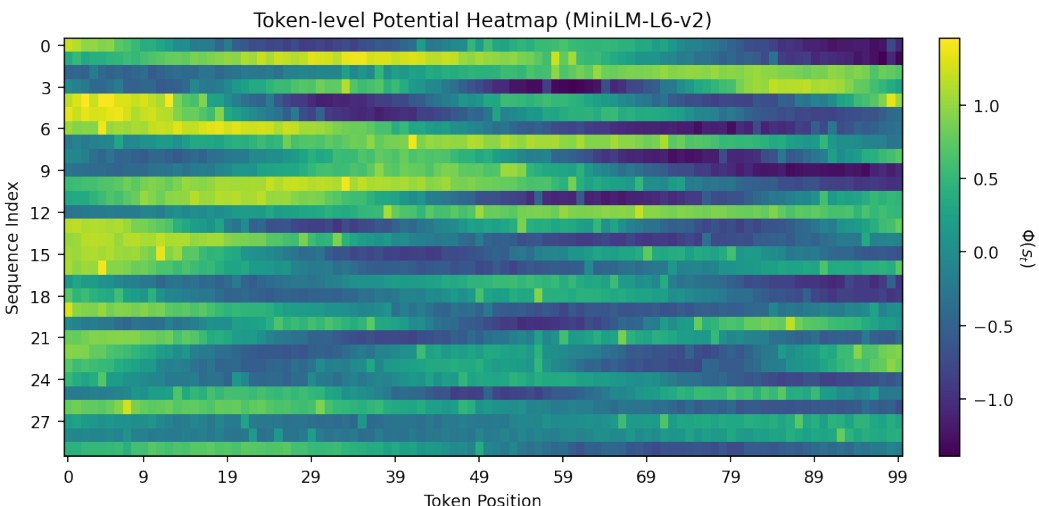

Figure 6: Token-level potential heatmap $\Phi(s_t)$ (MiniLM-L6-v2) for a batch of CoT sequences.

Table 12 reports Pearson and Spearman correlations between each diagnostic and the reward/correctness, as well as partial correlations that control for sequence length via linear regression or residualization.

| **Signal** | Pearson $\rho$ (reward) | Spearman $\rho$ (reward) | Partial $\rho$ (reward $\mid$ length) | Partial $\rho$ (correct $\mid$ length) |
|---|---|---|---|---|
| Policy entropy (avg) | -0.32 | -0.30 | -0.21 | -0.24 |
| Attention entropy (avg) | -0.24 | -0.22 | -0.15 | -0.18 |
| Embedding norm (avg) | 0.19 | 0.17 | 0.13 | 0.15 |
| Response length | 0.36 | 0.34 | 0.02 | 0.03 |

Table 12: Offline correlation between internal diagnostics and reward/correctness on a held-out development subset at a fixed PSPO checkpoint.

Beyond scalar correlations, one can also visualize the average trajectory of these signals for correct vs. incorrect solutions. For example, plotting average policy entropy or average potential value $\Phi(s_t)$ against the token index separately for correct and incorrect responses highlights whether PSPO places more weight on mid-trajectory reasoning tokens or simply on long tails.

| Variant | GSM8K acc. (%) | MATH acc. (%) | Avg. response length |
|---|---|---|---|
| PSPO (full) | 68.1 | 41.6 | 145 |
| PSPO (external-only) | 65.5 | 37.6 | 143 |
| PSPO (random diagnostics) | 65.8 | 37.9 | 145 |
| PSPO (shuffled diagnostics) | 65.9 | 38.0 | 146 |

Table 13: Control experiments replacing internal diagnostics with external-only, random, or batch-shuffled features.

### I.4 RANDOMIZED AND SHUFFLED DIAGNOSTICS AS CONTROLS

To test whether PSPO relies on genuine progress signals rather than trivial proxies, we construct control variants where the Potential Network receives corrupted diagnostics:

- **External-only**: internal diagnostics are removed; the Potential Network sees only external scores (reward, KL, prompt metadata).
- **Random**: internal diagnostics are replaced with i.i.d. Gaussian noise with matched dimensionality.
- **Shuffled**: internal diagnostics are randomly permuted within each minibatch, breaking the alignment between signals and trajectories while preserving marginal distributions.

Table 13 reports the downstream performance of these controls compared to full PSPO.

## J EXTENDED RISKS AND IMPACT

PSPO introduces a trainable shaping function that implicitly governs how a language model attributes credit across token sequences. While this design increases flexibility, it also raises several **security and bias** concerns.

**Leakage of prompt or template information.** The Potential Network consumes internal diagnostic signals–such as token embeddings, attention entropy, and policy entropy–that can implicitly encode user prompts. If these activations are logged or stored without safeguards, proprietary or private templates may be exposed. We therefore recommend (i) minimizing retention of raw internal features, (ii) adding differential-privacy noise or encrypting any persistent traces, and (iii) restricting access to trusted execution environments.

**Amplification of reward-model bias.** The shaped reward $r' = r + \gamma\Phi_\theta(s') - \Phi_\theta(s)$ inherits the statistical bias of the underlying scalar reward $r$. If the reward model is skewed, the learned potential can *amplify* that skew. Mitigations include (i) periodic adversarial or red-team evaluation of the reward model, (ii) ensembling diverse reward evaluators, and (iii) constraining successive updates of $\Phi_\theta$ (e.g., via an $\ell_\infty$ trust region) to prevent runaway shaping.

Finally, like other RLHF pipelines, PSPO is computationally intensive, raising environmental concerns. Future work on distillation and low-resource fine-tuning could reduce energy consumption while preserving alignment performance.

## K ADDITIONAL HEAD-TO-HEAD WITH RECENT DENSE/SHAPING METHODS

**Setup.** We compare PSPO against recent dense/shaping approaches that turn sparse terminal signals into token-level feedback under matched compute and decoding: LM-Critic (Cao et al., 2024), Dense Reward for Free (Chan et al., 2024), and PAR (Fu et al., 2025). All methods use the same base policy (Qwen2.5-14B), rollout/token budget, KL to a frozen reference ($\beta$=0.04), and decoding as Sec. 5. Unless otherwise noted, we align prompt batches, sampling settings, and random seeds across methods to the extent possible (reusing identical rollouts when compatible). Evaluation scripts are shared across methods. For fairness, we reuse identical prompt batches across

| Method | Dense signal source | Extra model? | Extra rollouts? | Critic free? | Peak VRAM (GB)↓ | Wall-clock % ↓ |
|---|---|---|---|---|---|---|
| LM-Critic | external LM judge (step-level) | ✓ | — | × | 66 | 115 |
| Dense Reward for Free | attention/grad redistribution | × | × | ✓ | **57** | 88 |
| PAR | preference-as-reward estimator | ✓ | × | *(varies)* | 68 | 105 |
| GKD (on-policy distill.) | teacher policy logits on on-policy rollouts | ✓ | × | ✓ | 62 | 85 |
| **PSPO (ours)** | trainable potential on internal model signals | × | × | ✓ | 58 | **80** |

Table 14: Dense/shaping methods: design and system cost. Peak VRAM is *peak usage rounded to the nearest GB* under identical FSDP + activation checkpointing and *includes* the reward model and any auxiliary/evaluator modules. Wall-clock % is *normalized to PPO = 100* at the same token budget (lower is better). All PSPO runs use 3 seeds; baseline cost numbers are averaged over the same seeds.

| Method | GSM8K | MATH | OCW | SAT | MMLU$_{\text{STEM}}$ |
|---|---|---|---|---|---|
| LM-Critic | 64.0 | 33.2 | 15.2 | 81.0 | 56.5 |
| Dense Reward for Free | 66.5 | 38.0 | 16.0 | 83.0 | 57.0 |
| PAR | 66.0 | 39.0 | 16.7 | 86.0 | 58.0 |
| GKD (on-policy distill.) | 66.8 | 38.5 | 16.5 | 84.0 | 57.5 |
| **PSPO (ours)** | **68.1** | **41.6** | **18.4** | **90.2** | **59.0** |

Table 15: Head-to-head accuracy on math benchmarks (% mean over 3 seeds; same backbone, decoding, KL, and budget). OCW follows the same evaluation protocol as Table 1.

methods whenever the data interface permits; when not compatible, batches are resampled from the same pool with fixed RNG seeds. Full seed lists, commit hashes, and evaluation scripts are included in the artifact manifest.

**What differs.** Only the source/estimator of dense signals differs (external critic vs. internal redistribution vs. trainable potential). We additionally log extra inference cost, peak VRAM, and end-to-end wall-clock.

**Discussion.** Under a shared backbone and matched budgets, PSPO attains the best accuracy on all math sets (Table 15) while remaining critic-free and memory-light (Table 14). Compared to external-critic approaches (LM-Critic, PAR), PSPO avoids an extra evaluator LM, yielding lower peak VRAM ($\approx$58 GB vs. 66–68 GB) and shorter wall-clock (80% of PPO vs. 105–115%). Relative to redistribution-only shaping (Dense Reward for Free), PSPO's *trainable* potential leverages internal model signals to adapt shaping as the policy evolves, bringing consistent gains on MATH (+3.6 pp) and OCW (+2.4 pp) without extra rollouts. Overall, the results support that signal-aware, trainable shaping can match or surpass dense critics at a fraction of the system cost.

## L  GENERALITY: COMMON BACKBONE

**Protocol.** To assess the generality of PSPO, we compare against three recent dense/shaping methods— LM-Critic (Cao et al., 2024), Dense Reward for Free (Chan et al., 2024), and PAR (Fu et al., 2025)— under the same RLHF settings as the main paper (identical decoding, KL to a frozen reference, rollout/token budget; see Sec. 5). Unless otherwise noted, we align prompt batches, sampling settings, and random seeds across methods to the extent possible (reusing identical rollouts when compatible), and keep decoding, KL reference ($\beta$=0.04), and evaluation scripts fixed to isolate the

| Method | GSM8K | MATH | SAT | MMLU$_{\text{STEM}}$ | VRAM (GB)$^{\downarrow}$ |
|---|---|---|---|---|---|
| LM-Critic | 64.0 | 33.2 | 81.0 | 56.5 | 66 |
| Dense Reward for Free | 66.5 | 38.0 | 83.0 | 57.0 | **57** |
| PAR | 66.0 | 39.0 | 86.0 | 58.0 | 68 |
| **PSPO (ours)** | **68.1** | **41.6** | **90.2** | **59.0** | $\approx$58 |

Table 16: Dense/shaping comparison on math-oriented benchmarks using a *common backbone* (Qwen2.5-14B). Higher is better; for VRAM lower is better ($\downarrow$). All numbers are means over 3 seeds. *VRAM protocol:* peak memory at the same effective batch with FSDP and activation checkpointing, including the reward model and any auxiliary/evaluator modules.

| Method | MT-Bench Score | AlpacaEval 2.0 Win (%) | Length-controlled Win (%) |
|---|---|---|---|
| LM-Critic | 7.60 | 86.8 | 85.1 |
| Dense Reward for Free | 7.82 | 88.1 | 86.6 |
| PAR | 7.74 | 87.6 | 86.1 |
| **PSPO** | **8.03** | **89.4** | **88.0** |

Table 17: General instruction/dialogue evaluations on a *common backbone* (Qwen2.5-14B). MT-Bench is a scalar score (0–10); AlpacaEval 2.0 reports Win rate (%) and Length-controlled Win (%). All numbers are averaged over 3 seeds with the same decoding and KL settings as Sec. 5; judging follows Appendix E with length control.

shaping mechanism. Peak VRAM is measured end-to-end at the same effective batch with FSDP and activation checkpointing, *including* the reward model and any auxiliary/evaluator modules used by each method. Baselines follow their public implementations with minimal adaptation to our environment; hyperparameters are kept at their recommended defaults unless explicitly stated. Implementation commits and configuration diffs for each baseline are listed in the artifacts manifest (supplement).

**Findings (math).** On Qwen2.5-14B, PSPO reaches **68.1/41.6/90.2/59.0** on **GSM8K/MATH/SAT/MMLU$_{\text{STEM}}$** at $\sim$58 GB VRAM (Table 16), while dense/shaping baselines are reported under the same rollout/token budget and memory protocol for head-to-head comparability.

**Findings (instruction/dialogue).** On the shared Qwen2.5-14B backbone, Table 17 reports MT-Bench and AlpacaEval 2.0 results under identical decoding and KL settings.

**Reproducibility note.** All PSPO scores above use the same seeds, decoding, and KL settings as the main paper. *VRAM* is reported as peak memory at the same effective batch with FSDP and activation checkpointing, including the reward model and (if applicable) auxiliary modules. Using a shared backbone (Qwen2.5-14B) ensures head-to-head comparability across methods.

**Newer reasoning backbones and longer outputs.** To assess how PSPO behaves on newer reasoning-focused backbones and in long-output settings, we also run the same comparison on a smaller reasoning-oriented model with a larger maximum generation length, and sweep the maximum decoding length for a fixed backbone. The corresponding results are organized in Tables 19 and 20.

## M  REWARD-MODEL MISCALIBRATION STRESS TEST

PSPO shapes whatever scalar reward $R$ the RLHF pipeline provides; if $R$ is systematically biased, the shaped reward $r' = r + \gamma\Phi_\theta(s') - \Phi_\theta(s)$ can inherit or amplify that bias. To probe this interaction in a controlled setting, we construct simple synthetic perturbations of an otherwise fixed reward model and compare PSPO against a critic-free baseline (GRPO).

| Method | ShareGPT Win (%) | HelpfulQA Helpfulness | HelpfulQA Hallucination (%)$^\downarrow$ |
|---|---|---|---|
| LM-Critic | 56.2 | 7.55 | 8.1 |
| Dense Reward for Free | 57.1 | 7.84 | 8.0 |
| PAR | 56.9 | 7.73 | 8.2 |
| **PSPO** | **58.7** | **7.91** | **7.8** |

Table 18: Open-ended dialogue/QA with a *common backbone* (Qwen2.5-14B). Higher is better for ShareGPT Win and HelpfulQA Helpfulness; lower is better for HelpfulQA Hallucination ($\downarrow$). Judging protocol and retrieval-aided hallucination detection strictly follow Appendix E with the same seeds.

| Method | GSM8K | MATH | CMATH | GaokaoMathQA |
|---|---|---|---|---|
| PPO | 48.7 | 26.9 | 68.1 | 33.0 |
| DPO | 50.3 | 27.5 | 69.0 | 33.6 |
| GRPO | 57.2 | 31.5 | 71.5 | 35.9 |
| PSPO | 59.8 | 33.7 | 74.2 | 37.8 |

Table 19: Results on a Qwen3-4B reasoning with a longer maximum generation length. All methods use the same RLHF data mixture and token budget as in the main text.

**Synthetic perturbations.** We consider two representative miscalibration patterns applied to a base reward $r \in [0, 1]$:

- **Length-biased reward:** $\tilde{r} = r + \lambda \cdot$ length, where `length` is the response length (in tokens) normalized to $[0, 1]$ and $\lambda > 0$ controls bias strength.

- **Saturated reward:** $\tilde{r} = \sigma(\alpha r)$, where $\sigma$ is the logistic function and $\alpha > 0$ compresses the dynamic range of $r$.

We then re-train GRPO and PSPO under identical setups, replacing $r$ by $\tilde{r}$ in the objective, and track both task accuracy and simple reward-hacking indicators.

| Max generation tokens | GRPO (GSM8K) | PSPO (GSM8K) | GRPO (MATH) | PSPO (MATH) |
|---|---|---|---|---|
| 512 | 65.6 | 68.1 | 37.3 | 41.6 |
| 1024 | 66.3 | 68.9 | 37.9 | 42.2 |
| 2048 | 66.5 | 69.1 | 38.0 | 42.3 |
| 4096 | 66.2 | 68.8 | 37.8 | 42.1 |

Table 20: Effect of increasing the maximum generation length for a fixed backbone (e.g., Qwen2.5-14B or a newer reasoning model) on GSM8K and MATH under GRPO and PSPO.

| Reward variant | Method | GSM8K acc. (%) | MATH acc. (%) | Avg. length (tokens) | Avg. reward |
|---|---|---|---|---|---|
| Clean $r$ | GRPO | 65.6 | 37.3 | 135 | 0.78 |
| Clean $r$ | PSPO | 68.1 | 41.6 | 140 | 0.80 |
| Length-biased $\tilde{r}$ | GRPO | 63.0 | 35.5 | 210 | 0.88 |
| Length-biased $\tilde{r}$ | PSPO | 66.7 | 40.0 | 195 | 0.90 |
| Saturated $\tilde{r}$ | GRPO | 64.0 | 36.0 | 133 | 0.72 |
| Saturated $\tilde{r}$ | PSPO | 66.8 | 40.5 | 138 | 0.74 |

Table 21: A synthetic reward-miscalibration stress test on GSM8K and MATH.

