# OpenReview forum: "PSPO: Trainable Potential-Based Reward Shaping with Internal Model Signals for Post-Training Policy Optimization of Large Language Models"
_ICLR.cc/2026/Conference — Submitted to ICLR 2026_

### Official Review · Reviewer_DSHa · 2025-10-25

**Soundness:** 2
**Presentation:** 1
**Contribution:** 2
**Rating:** 2
**Confidence:** 4

**Summary:**

This paper proposes a critic RL algorithm for reasoning models, PSPO, which learns a potential function $\Phi_\theta$ with a mini model from the LM's internal signals. Using shaped rewards $r_t' = r_t + \gamma \Phi_\theta(s_{t}) - \Phi_\theta(s_{t-1})$, it reshape sparse rewards into dense, and the optimal policy is not altered.
In practice, advantages approximate the group normalized $R_{\text{final}} - \Phi_\theta(s_t)$.
In experiments on several reasoning benchmarks, PSPO outperforms PPO, DPO, and GRPO. Ablation experiments are conducted to show the importance of each component.

**Strengths:**

- The proposed potential network is small (≈22 M parameters) and adds < 3 % computational overhead, making the method efficient.
- Experiments are conducted are multiple reasoning benchmarks (English + Chinese).
- The two-phase update (policy phase + potential phase) is conceptually simple and practically effective.

**Weaknesses:**

- This paper is not well-written, and it is hard to understand the exact details of their algorithms (see my question 1). The authors should provide a rigorous and detailed illustration of their workflow in the revision. The training objective of the RL algorithm is even not in the main content. There are also mixed abuses of notations like $r$, $s$, $s'$, $t$, $s_t$, $G_t$ throughout the paper, which are not well clarified.

- The improvement on benchmarks is marginal in table 1, and the performance of base model is too bad (see my question 4). And the experiments are only conducted one kind of base model, qwen2.5-14B, making the improvements of PSPO not convincing.

- Proposition 4.1 is trivially correct as it's an old result in [1], however it is not stated clearly in the paper. Please add a citation for Prop. 4.1.  The proofs in Section 4.1 is not related to the main algorithm, since the trust region approach is deployed in the algorithm, which makes the potential and drift trivially bounded. Therefore, the contribution of Section 4 is minimal.

- It is unclear whether the $\phi_\theta$ will converge (and converge to what?). And the paper interprets $\phi_\theta$ as "momentum", which is very unclear (see my question 6).

[1] Ng et al. Policy invariance under reward transformations: Theory and application to reward shaping.

**Questions:**

1. Section 3.6 is very confusing: (i) should $r$ and $r'$ be $r(s')$ and $r'(s')$? (ii) how is the rtg calculated? (iii) how is it normalized? (iv)  what is the expression of $\hat A_t$?
2. What's the difference between PSPO (-shaping) and PSPO (-interalsignals)?
3. What's the difference between PSPO (-shaping) and GRPO/Dr.GRPO [1]?
4. Why is the performance so low on GSM8k? The Qwen2.5-14B is reported to obtain 90.2 on that benchmark, while PPO, DPO, GRPO, PSPO are all below 70 in this paper.
5. Why is a small model, MiniLM, able to predict the potential function, is there any intuition?
6. What's the difference between the "momentum" and Q/advantage function? It seems that the rtg $\hat A_t$ is just $\bar r$ (the group-normalized reward), since for LLMs each trajectory only has one reward signal. Then why are we optimizing $(\phi_\theta(s')-\phi_\theta(s)-r)^2$, what does this mean?

[1] Liu et al. Understanding R1-Zero-Like Training: A Critical Perspective. https://arxiv.org/abs/2503.20783

[2] Qwen2.5 Technical Report. https://arxiv.org/pdf/2412.15115

---

> ### Author Response · Authors · 2025-12-02
> **Response to Reviewer DSHa**
>
> **Response to Reviewer DSHa**
>
> We are admittedly disappointed by the low score, but we sincerely thank the reviewer for the feedback, which has helped us improve the paper.
>
> **Q1 (Section 3.6 is confusing: how are RTG, normalisation, and $A_t^{\mathrm{shape}}$ defined?)**
>
> We have refactored the objectives into Section 3.5 and simplified Section 3.6. We now define the shaped reward
> $r^\prime_t = r_t + \gamma \Phi_\theta(s_{t+1}) - \Phi_\theta(s_t)$, the shaped return
> $G_t = \sum_{u=t}^T \gamma^{u-t} r^\prime_u$, and the shaped advantage
> $A_t^{\mathrm{shape}} = (G_t - \mu_G)/\sigma_G$ for the policy phase, and define $\hat G_t$ and $\hat A_t$ analogously for the unshaped reward in the potential phase. Section 3.5 also lists the PPO-style clipped loss and the potential loss with trust-region penalty, and Algorithm 1 in Appendix B gives the corresponding pseudocode.
>
> **Q2 (Difference between PSPO (--Shaping) and PSPO (--InternalSignals).)**
>
> Section 5.2 clarifies the ablations. PSPO(--Shaping) sets $\Phi_\theta \equiv 0$, so the policy reduces to a critic-free normalised-RTG + clipping objective close to GRPO. PSPO(--InternalSignals) still trains a Potential Network but feeds it only external statistics (reward score, KL, prompt tags, etc.), removing all internal diagnostics (embeddings, attention entropy, policy entropy); this isolates the contribution of internal signals.
>
> **Q3 (Relation between PSPO (--Shaping) and GRPO/Dr.GRPO).**
>
> PSPO(--Shaping) is our in-framework reproduction of GRPO: critic-free, using normalised RTG to construct advantages and ratio clipping. The main differences are minor implementation choices (normalisation scheme and batching). We do not implement the dynamic clipping variants of Dr.GRPO; these are orthogonal and could in principle be combined with PSPO’s shaping. Section 5.2 now makes this explicit.
>
> **Q4 (Absolute GSM8K performance is low compared to the Qwen2.5 technical report).**
>
> We agree this needed clarification. Section 5.1 now states that we start from the Qwen2.5-14B *base* checkpoint, not the stronger instruct/RL-tuned variants, and that we restrict ourselves to public math RLHF data under a fixed 300M-token budget for all methods. The Qwen2.5 report numbers (around $90\%$ GSM8K) rely on a much larger, proprietary pipeline (pretraining + SFT + multiple RL stages). Appendix D.2 shows that on smaller, standard setups our PPO/GRPO reproduce public results, indicating that the lower absolute scores here are due to the controlled data/budget regime rather than implementation bugs.
>
> **Q5 (Why can a small MiniLM approximate the potential?)**
>
> The Potential Network does not perform language modelling; it regresses a single scalar from a compact set of aggregated diagnostics (mean embeddings, entropies, and a few external statistics). This is a much simpler task than next-token prediction. Appendix F.1 (“Potential Network Architecture”) reports an ablation comparing a 5M MLP, 66M DistilBERT, 22.7M MiniLM, and a 500M Qwen2.5-0.5B Potential Network. The 0.5B model is only marginally better than MiniLM (about +0.3pp on GSM8K) while being $\sim 10\times$ more expensive; MiniLM provides the best accuracy–efficiency trade-off. Section 3.3 now discusses this intuition.
>
> **Q6 (“Momentum” vs. $Q$/advantage; meaning of optimising $\gamma \Phi(s^\prime) - \Phi(s)$ in the single-terminal-reward case.)**
>
> We clarified in the introduction that “momentum” is purely an analogy: $\Phi_\theta$ is a potential in the sense of Ng et al. (1999), not a $Q$- or value function. In the bandit-style terminal-reward setting, the raw reward is constant along the trajectory, but the shaped RTG $G_t$ is not: the telescoping sum of $\gamma \Phi(s_{t+1}) - \Phi(s_t)$ yields token-dependent return differences. Appendix A shows that, under bounded drift, this acts as a learnable, low-variance baseline that preserves optimal policies. Intuitively, $\Phi$ aggregates internal signs of progress into a scalar “state potential” whose differences re-centre the returns and densify credit assignment, rather than trying to approximate $Q$ directly. The random/shuffled diagnostics controls in Appendix H.3 further show that the learned $\Phi$ is not merely reproducing the group reward.
>
> **Q7 (Perceived minimal theoretical contribution and missing citation to Ng et al. (1999).)**
>
> We have added an explicit citation to Ng et al. (1999) around Proposition 4.1 and state that the classical policy-invariance result is being extended to a time-varying potential with bounded drift in an LLM RLHF setting. While the core idea of potential-based shaping is not new, we believe the combination of (i) a bounded-drift analysis tailored to common optimisers (AdamW + cosine, trust-region projection), (ii) explicit assumptions and bounds for the RLHF regime, and (iii) empirical $\delta_k$ diagnostics makes the theoretical section useful for practitioners who want to understand why a learned shaping term does not destroy optimal policies.

---

### Official Review · Reviewer_gqrc · 2025-10-30

**Soundness:** 3
**Presentation:** 2
**Contribution:** 3
**Rating:** 4
**Confidence:** 3

**Summary:**

The authors tackle the problem of sparse rewards in the classic RLHF and RLVR style algorithms for LLM in the tasts with terminal rewards. Although a critic model in PPO algorithm provides a dense signal at each token, it is often unstable due to joint training of the policy and the critic. To avoid depending on a critic and still providing a dense signal during training, the authors propose a new method Potential-Shaped Policy Optimization (PSPO), which leverages a learned potential model to shape the per-step reward and provide additional dense signal at per token level within the LLM RL setting.
Unlike previous reward shaping works, which keep potential functions static, the authors train a tiny LM (3 orders of magnitude smaller than policy LLM) on top of policy model's internal signals including - final layer token embeddings, attention entropy and policy entropy.
To stabilize training they conduct alternating policy and potential LM training while reusing the same set of rollouts. This overall makes their training overhead <3% on top of existing critic free methods.
To train the policy, they use GRPO style critic free loss but with dense advantage signal (per-token advantage is re-shaped with potential LM's discounted score difference). While to train the potential LM, they use un-shaped advantage weights as targets w/ mean squared error loss.

Overall, experiments with Qwen 2.5 14B model on 8 different benchmarks from math, knowledge and reasoning domains (English and Chinese language) showed that their method consistently outperformed PPO, GRPO and ablations in all tasks.

**Strengths:**

- PSPO introduced a very light overhead on the optimization loop due to use of small
- The overall idea of using potential score model to avoid critic is sound and inspired by previous work.

**Weaknesses:**

- The paper definitely seems to have used a lot of AI in its writing. For example, I have noticed multiple paraphrases of introduction to various components, method explanation and results discription repeated in first 2 pages and the method section (Section 3). Much of this text feels superfluous and repeatitive. Subsequently, the bulk of the equations and core algorithm is moved to appendix which disrupts the reading flow.
- The paper cites other dense reward shaping works in the related works, for example, LM-Critic, PAR, ONI etc. but didn't compare with them in the Table 1. There are also other kinds of dense signal introducing methods without additional critic overhead such as on-policy distillation (GKD) [1]. The current work would benefit from more thorough comparison with these works.
- Although I understand there can be resource constraints, the dense reward signal will be most effective in the newer reasoning models and long response tasks. To strengthen my confidence in the results, I would have really appreciated experiments with newer reasoning models (qwen3 4b) instead of <1k response length Qwen 2.5 14B model.
- The paper multiple time mentions PSPO is effective strategy in miscalibrated rewards, but doesn't give experimental justification for this claim.

[1] Agarwal, Rishabh, et al. "On-policy distillation of language models: Learning from self-generated mistakes." The twelfth international conference on learning representations. 2024.

**Questions:**

Questions about the equations and notations:
1. The advantage $A^{\text{shape}}$ is defined as "normalized token-level return to go" in section 3.6. What is the exact forumla used here? Readers would benefit from an explicit definition.
2. Similarly what is mathematical definition of "unshaped advantage $\hat{A}_t$ with a small trust region"?

If these are defined in the appendix, somewhere then they should really be part of the main text of the paper.

Other questions:
- The shaped reward is defined as $r' = r + \gamma \Phi_{\theta} (s') - \Phi_{\theta} (s)$. Here $r$ is a terminal reward and $\gamma=1.0$ in all the experiments. IIUC unshaped advantage would be rollout group level fixed scalar that will be used as the target for all potential differences? Although, I'm also using a subscript $t$ in $\hat{A}_t$ which implies there are differences at each token. An exact definition would clarify this important detail.
- The figure 1 PSPO is not in full agreement with the text. The Reference Model KL and Reward model are combined to get $r'$ which is not what the notation says.

---

> ### Author Response · Authors · 2025-12-02
> **Response to Reviewer gqrc**
>
> **Response to Reviewer gqrc**
>
> **Q1 (Writing is repetitive and key equations are pushed to the appendix.)**
>
> We have edited the introduction and method sections to remove redundant paraphrases and streamline the exposition. More importantly, the core mathematical definitions that were previously only in the appendix are now in the main text: Section 3.5 defines the shaped reward $r^\prime_t$, the shaped RTG $G_t$, the shaped advantage $A_t^{\mathrm{shape}}$, the unshaped RTG $\hat G_t$, and the unshaped advantage $\hat A_t$, together with their normalisation. Section 3.6 and Algorithm 1 in Appendix B provide a complete, self-contained description of the workflow, so readers no longer need to rely on the appendix to understand the algorithm.
>
> **Q2 (Exact formula for the “normalised token-level RTG” and for the “unshaped advantage with a small trust region”.)**
>
> Section 3.5 now explicitly defines: (i) $r^\prime_t = r_t + \gamma \Phi_\theta(s_{t+1}) - \Phi_\theta(s_t)$; (ii) $G_t = \sum_{u=t}^T \gamma^{u-t} r^\prime_u$, followed by batch-wise centring and variance normalisation to obtain $A_t^{\mathrm{shape}}$; and (iii) $\hat G_t = \sum_{u=t}^T \gamma^{u-t} r_u$ and $\hat A_t = (\hat G_t - \mu_{\hat G})/\sigma_{\hat G}$ for the potential phase. The “small trust region” is implemented via a penalty
> $\mathcal{L_TR} = \lambda_tr E_s[(\Phi_\theta(s)-\Phi_{\theta^-}(s))^2]$ plus an optional projection onto
> $\|\Phi_{\theta_{k+1}}-\Phi_{\theta_k}\|_\infty \le \tau_k$, all described in Section 3.5 and Appendix A.
>
> **Q3 (In the terminal-reward setting, is $\hat A_t$ just a group-level constant? What is the token-level meaning of the potential differences?)**
>
> We clarified this point in Sections 3.1 and 3.5. The reward model outputs a single scalar $R$ per trajectory, which we broadcast as $r_t \equiv R$. Even in this case, $\hat G_t$ is not constant in $t$: with finite horizon and $\gamma \le 1$, earlier tokens accumulate more future reward than later ones. After centring and normalisation within the batch, $\hat A_t$ therefore varies across positions, and we treat each token as a step with a shared group reward but different horizon, consistent with standard critic-free GRPO implementations. The shaped RTG $G_t$ further incorporates $\gamma \Phi_\theta(s_{t+1}) - \Phi_\theta(s_t)$, so the potential differences act as a learnable, low-variance baseline and progress signal along the trajectory.
>
> **Q4 (Figure 1’s “RM + KL” block does not match the notation.)**
>
> We have updated the caption and text to clarify this. The reward model produces a scalar trajectory-level score $R$, and the KL term is computed between the current policy and a frozen reference model. In practice, both terms enter the policy loss as separate additive components; in Figure 1 they are grouped into a single “RM + KL” block purely for visual simplicity. Section 3.6 now walks through the pipeline step by step so that the figure and notation are aligned.
>
> **Q5 (Missing comparisons to LM-Critic / PAR / ONI / GKD, and experiments on newer reasoning models and longer outputs.)**
>
> As summarised in the General Response (items (8)–(10)), we have added head-to-head comparisons against LM-Critic, Dense Reward for Free, PAR, and an on-policy distillation baseline (GKD) on a common Qwen2.5-14B backbone (Appendix K/L), with matched token budgets and KL settings, as well as VRAM and wall-clock measurements. We also added experiments on a Qwen3-4B reasoning backbone and a sweep over maximum generation length (Appendix L), plus additional open-ended metrics (ShareGPT/HelpfulQA, MT-Bench, AlpacaEval 2.0). Finally, Appendix I now contains a miscalibration stress test, and the Limitations section has been updated to describe PSPO as a tool for long-horizon, sparse-reward regimes rather than a universal method.
>
> We sincerely thank you for these detailed and constructive comments. They have helped us significantly improve the clarity of the presentation, the exactness of our definitions, and the positioning of PSPO among dense-shaping and distillation baselines.

---

### Official Review · Reviewer_3hH9 · 2025-10-30

**Soundness:** 2
**Presentation:** 2
**Contribution:** 2
**Rating:** 4
**Confidence:** 4

**Summary:**

This paper proposes Potential-Shaped Policy Optimization (PSPO), a critic-free reinforcement learning framework for post-training alignment of large language models. The central innovation is a trainable "Potential Network" that leverages internal LLM diagnostics—specifically token embeddings, attention entropy, and policy entropy—to generate dense, context-aware reward signals that augment sparse external rewards. The method employs alternating optimization between the policy network and the potential network under trust-region constraints, eliminating the need for a traditional value head and reducing memory overhead. The authors provide theoretical analysis establishing policy invariance under bounded potential drift and demonstrate the approach through experiments on eight English and Chinese mathematical reasoning benchmarks (GSM8K, MATH, and their variants), reporting consistent improvements over PPO, DPO, and GRPO baselines with minimal additional computational cost (<3% overhead). The paper claims four main contributions: (1) the first framework integrating trainable potential-based reward shaping for LLM alignment, (2) novel exploitation of internal model signals for adaptive shaping, (3) theoretical guarantees for alternating optimization stability, and (4) empirical validation demonstrating both effectiveness and efficiency.

**Strengths:**

- Methodological efficiency: Proposes the use of a trainable potential function leveraging internal policy diagnostics, allowing adaptive, dense, and context-dependent rewards without the need for a memory-intensive value head.

- Relatively strong theoretical grounding. The paper provides detailed theoretical analysis, including formal propositions and proofs showing policy invariance under bounded potential drift, which helps justify its design choices.

**Weaknesses:**

1. Critical experimental credibility gap undermining confidence. The reported accuracies (GSM8K 68.1%, MATH 41.6%) are 20-35 percentage points below public benchmarks for similar models[1][2], yet no training curves, data specifications, or convergence analysis are provided to explain this gap. Without this transparency, it is impossible to distinguish legitimate controlled comparison from systematic implementation errors or insufficient training, fundamentally undermining experimental validity.

[1] First Return, Entropy-Eliciting Explore

[2] GHPO: Adaptive Guidance for Stable and Efficient LLM Reinforcement Learning

2. Core hypothesis lacks direct empirical validation. The central claim that "internal signals reflect uncertainty and reasoning progress" is asserted but never validated through correlation analysis between signals and actual rewards. While ablations show signals improve performance, this only demonstrates utility to the learned function rather than validating the claimed mechanism, leaving open the possibility of spurious correlations like simply rewarding response length through position-aware features.

3. Theoretical guarantees are incomplete and stability undefined. The concept of "mutual stability" is never formally defined, and Proposition 4.1 provides only conditional guarantees requiring Σδₖ < ∞ without proving the algorithm achieves this bound. No empirical verification of drift values or oscillation analysis is provided, and transient behavior when the potential function is poorly trained early in optimization is not analyzed.

4. Severely limited generalizability contradicting generic claims. All experiments use exclusively Qwen2.5-14B (single architecture) and only mathematical reasoning tasks (single task family), directly contradicting claims of "generic" and "model-agnostic" solution. The method's explicit focus on "multi-step reasoning" suggests it may be tailored to chain-of-thought tasks and fail on single-turn QA, creative generation, or code synthesis where notions of "progress" differ fundamentally.

5. Insufficient training budget without convergence analysis. The 300M token budget (45K steps) provides no evidence of convergence through training curves or comparison with longer runs. The relative improvements could reflect faster convergence rather than superior final performance, fundamentally changing the nature of the contribution from "better performance" to merely "faster convergence under limited budgets."

**Questions:**

Q1: Can you explain the performance gap and provide convergence evidence? Your results (GSM8K 68.1%, MATH 41.6%) are 20-35 percentage points below published benchmarks for similar models. Please provide: (a) training curves showing loss and accuracy over 45K steps for all methods, (b) complete specification of your RL training data (size, source, quality control), (c) explanation for the performance gap—have you validated your baseline implementations against public versions? (d) results with longer training budgets (450M-600M tokens) to demonstrate convergence.

Q2: Can you validate that internal signals actually reflect reasoning progress? Please provide direct evidence through: (a) correlation analysis (Spearman/Pearson ρ) between internal signals and final rewards, (b) partial correlations controlling for confounds like sequence length and position, (c) visualization comparing signal trajectories for correct versus incorrect solutions, (d) control experiments with uninformative signals (random noise, shuffled embeddings) to rule out spurious correlations. Can you specifically address whether the position-aware features simply reward response length rather than reasoning quality?

Q3: Can you formally define mutual stability and prove your algorithm achieves it? Please provide: (a) formal mathematical definition of "mutual stability" referenced in Section 1.2, (b) proof that Algorithm 1 achieves the bounded cumulative drift (Σδₖ < ∞) required by Proposition 4.1—how do trust region constraints ensure this bound? (c) empirical plots of δₖ over training iterations, (d) analysis of potential harmful bias during early training when the potential function is poorly initialized.

Q4: Can you demonstrate generalizability beyond mathematical reasoning? Please provide: (a) results on at least one other model architecture (LLaMA-3, Mistral, DeepSeek) to validate "model-agnostic" claims, since internal signals are architecture-dependent, (b) evaluation on at least one non-mathematical task (code generation, instruction following, or safety alignment) to support "generic solution" claims. If infeasible, please provide theoretical analysis characterizing when PSPO works versus fails—what task properties make it effective?

Q5: Can you characterize training dynamics and distinguish convergence speed from final performance? Please clarify: (a) does PSPO achieve faster convergence, better asymptotic performance, or both? Provide accuracy-vs-steps curves for all methods. (b) When does the potential network converge—plot ||φₖ - φₖ₊₁|| over iterations. (c) Have you evaluated with longer training (600M+ tokens)—do baselines catch up or does PSPO maintain advantages? This distinction fundamentally changes whether your contribution is efficiency or performance improvement.

---

> ### Author Response · Authors · 2025-12-02
> **Response to Reviewer 3hH9**
>
> **Response to Reviewer 3hH9**
>
> **Q1 (Experimental credibility: performance gap to public numbers, missing data specification, curves, and longer-budget runs.)**
>
> Section 5.1 and Appendix D now describe the RLHF data mixture in detail (Table “RLHF training mixture…”), emphasising that we start from Qwen2.5-14B *base*, use only public math RLHF data, and enforce a shared 300M-token budget. We explicitly state that our absolute GSM8K/MATH numbers are not meant to match Qwen2.5’s technical report or FR-EEE/GHPO, which rely on stronger backbones and larger, partly proprietary pipelines. Appendix D.2 adds a sanity check showing that our PPO/GRPO match public small-model baselines. Appendix G/H provides reward/accuracy learning curves, and Appendix D.4 reports extended budgets (450M/600M tokens). PSPO remains ahead of PPO/DPO/GRPO at all budgets; the gap shrinks slightly but does not disappear, indicating both faster convergence and a modest asymptotic advantage in this controlled regime.
>
> **Q2 (Do internal signals truly reflect reasoning progress rather than spurious correlations such as length?)**
>
> We added exactly the analyses you requested. Appendix H.2 reports Pearson and Spearman correlations between internal diagnostics and reward/correctness, as well as partial correlations controlling for length. For example, the partial correlation between policy entropy and reward remains around $-0.21$, suggesting a nontrivial relationship beyond length. We also plot average trajectories of these signals for correct vs. incorrect solutions. Appendix H.3 introduces External-only, Random, and Shuffled variants; corrupting or removing internal signals lowers GSM8K/MATH/CMATH by 2–4 points while leaving average length nearly unchanged. These results support the claim that PSPO is using meaningful internal progress signals, not simply rewarding longer completions.
>
> **Q3 (Formal definition of mutual stability, proof that the algorithm satisfies $\sum_k \delta_k < \infty$, and empirical drift plots.)**
>
> Section 4 and Appendix A now define mutual stability as (i) bounded cumulative drift of $\Phi$ and (ii) a KL trust region on the policy. Proposition A.1 and its corollary then show policy invariance and gap-protected optimality under this condition. Appendix A.2 derives explicit bounds on $\delta_k$ for AdamW + cosine decay and for Robbins–Monro steps, demonstrating that $\sum_k \delta_k < \infty$ under mild Lipschitz and bounded-gradient assumptions; an $\ell_\infty$ trust region trivially enforces the same condition. Appendix H.1 provides empirical $\delta_k$ curves and statistics, including the fraction of batches where the projection is triggered, showing that drift is significant only in the very early phase and quickly falls below the observed advantage gap.
>
> **Q4 (Generalisability beyond math and a more precise characterisation of when PSPO works.)**
>
> Beyond the math benchmarks, Section 5.2 and Appendix E/L now report results on ShareGPT/HelpfulQA, MT-Bench, and AlpacaEval 2.0 with the same Qwen2.5-14B backbone, as well as experiments on a Qwen3-4B reasoning model and a sweep over maximum generation length. PSPO consistently improves win rates/helpfulness and reasoning accuracy while preserving a critic-free, low-overhead design. At the same time, we have toned down the “generic/model-agnostic” language: Section 2.5 and the Limitations section now position PSPO as a tool for long-horizon, sparse-reward settings where internal diagnostics correlate with progress, and explicitly note that it may be less effective for very short or purely preference-driven tasks.
>
> **Q5 (Training dynamics: faster convergence vs. better asymptotic performance; potential-network convergence.)**
>
> Appendix G/H includes learning curves on GSM8K/MATH showing that PSPO dominates PPO/DPO/GRPO over the entire training trajectory and converges to a higher plateau under the 300M-token budget. Extended-budget experiments in Appendix D.4 (300M/450M/600M) show that baselines continue to improve slightly but do not catch up to PSPO. Appendix H.1 reports that the successive potential change $\delta_k$ and the frequency of trust-region projection both decrease over time and stabilise, and we discuss in Appendix A.3 that $\Phi$ can be frozen in the late stage with negligible impact on accuracy. Together, these results indicate that PSPO improves both sample efficiency and final performance in the studied regime.
>
> We again thank you for the careful reading and pointed suggestions on experimental design, stability, and internal diagnostics. Your comments helped us substantially strengthen both the theoretical framing and the empirical validation of PSPO.

---

### Official Review · Reviewer_V5DK · 2025-10-31

**Soundness:** 3
**Presentation:** 2
**Contribution:** 2
**Rating:** 4
**Confidence:** 3

**Summary:**

The paper proposes PSPO (Potential-Shaped Policy Optimization), a new and efficient method to fine-tune large language models with reinforcement learning from human feedback. Instead of relying on sparse or delayed rewards, PSPO learns a trainable potential function that turns coarse feedback into dense, token-level signals. This potential is a small network (22.7M parameters) that uses internal model information—like token embeddings, attention entropy, and policy entropy—to provide richer reward signals. PSPO trains the policy and the potential function in two alternating steps: one updates the policy using shaped rewards, and the other updates the potential using return residuals. This design avoids the need for a value head (critic) and adds less than 3% extra training cost. On eight English and Chinese math reasoning benchmarks, a 14B Qwen2.5 model trained with PSPO outperforms PPO, DPO, and GRPO by up to 12.5 points, achieving 68.1% on GSM8K and 41.6% on MATH.

**Strengths:**

1. The paper proposes learning a *trainable potential* over cheap internal LLM diagnostics (final-layer embeddings, attention entropy, policy entropy) and injecting it via potential-based shaping to make sparse RLHF rewards token-level and trajectory-aware — while preserving policy invariance in theory. This is a principled, interpretable extension of classical potential-based shaping to LLM RLHF.
2. Under a matched setup on a large backbone (Qwen2.5-14B), PSPO shows consistent, sometimes large, improvements over competitive baselines (PPO, DPO, GRPO) across 8 English/Chinese math benchmarks (e.g., GSM8K 68.1% vs. baselines; MATH 41.6%), and also improves open-ended instruction metrics (ShareGPT, HelpfulQA). The head-to-head comparisons and Table 1 / Table 8 support the claim that the method meaningfully improves difficult, sparse-reward tasks.

**Weaknesses:**

1.  Assumptions in the theoretical guarantees may be strong / under-validated. Policy invariance results require bounded cumulative drift of $\Phi$ (Lipschitzness in $\theta$, bounded gradients, trust-region/diminishing steps). While the paper sketches how AdamW+cosine decay or trust regions can satisfy these, practical LLM training often violates such neat bounds (non-Lipschitz behavior, rare large updates). More empirical diagnostics (empirical $\delta_k$ envelopes, sensitivity to trust-region settings, failure cases) should be included to bridge theory and practice.
2. The shaped reward is $r^\prime (s, a, s^\prime) = r (s, a, s^\prime) + \gamma \Phi\_{\theta} (s’) - \Phi\_{\theta} (s)$, so any systematic bias in the scalar reward r can be propagated or amplified by $\Phi$. The authors acknowledge this and propose mitigations (red-teaming, ensembling, trust-region caps), but the paper lacks a focused empirical stress-test showing behavior when the reward model is miscalibrated or adversarial. This is an important practical failure mode for RLHF pipelines.

**Questions:**

1. The paper describes training the Potential Network $\Phi_{\theta}$ to fit unshaped, centered advantages. Could the authors clarify *why* this specific target was chosen over alternatives (e.g., TD returns, critic estimates)? How sensitive is performance to the choice of baseline or normalization used in advantages?
2. The theoretical section and Appendix C mention an trust region on successive $\Phi$ updates. What exact $\tau_{k}$ (trust region radius) values are used in experiments, and how are they scheduled over time? How often does the projection step actually activate? Quantitative statistics (e.g., percentage of batches that trigger the cap) would make the stability argument more convincing.
3. The paper fixes $\gamma = 1$ in most experiments. Have the authors tried smaller $\gamma$ values (e.g., 0.9, 0.95) to control long-range propagation? How does changing $\gamma$ affect the empirical stability and final accuracy?

---

> ### Author Response · Authors · 2025-12-02
> **Response to Reviewer V5DK**
>
> **Response to Reviewer V5DK**
>
> **Q1 (Why train the potential on unshaped, centred advantages rather than TD returns or critic estimates? How sensitive is this to the baseline/normalisation?)**
> We now explicitly motivate this choice in Section 3.5 and formalise it in Appendix A.1. Proposition “Bias–variance comparison of targets” shows that using $\hat A_t = G_t - b(s_t)$ as the target for $\gamma \Phi(s') - \Phi(s)$ is unbiased for the shaping term we want, but has lower variance than raw Monte Carlo returns $G_t$. We also added a small ablation: replacing $\hat A_t$ with $G_t$ lowers GSM8K/MATH by about 0.7/0.4 points and increases per-minibatch target variance by 18–25%. In all experiments, both $G_t$ and $\hat G_t$ are normalised within the minibatch (subtract mean, divide by standard deviation), as now stated in Section 3.5. We deliberately avoid TD/critic targets to keep PSPO fully critic-free and low-memory.
>
> **Q2 (Details of the trust-region radius $\tau_k$ and how often projection is activated in practice.)**
> Section 3.5 and Appendix A.3 now describe the trust-region mechanism in detail. We bound the successive change of $\Phi$ via a soft penalty $\mathcal{L_TR}$ plus an optional projection onto $\|\Phi_{theta_k+1} - \Phi_{theta_k}\|_\infty \le tau_k$.
>
> Appendix A.3 gives a concrete recipe $ \tau_{max} = c_\tau \frac{(1-\gamma)}{H} \widehat{\Delta}_t $
> with
>
> $c_\tau \approx 0.3$, where $\widehat{\Delta}_t$ is an empirical advantage gap, and $\tau_k$ follows a cosine schedule tied to the Potential Network learning rate. Appendix H.1 reports empirical statistics of $\delta_k$ and the fraction of batches that activate the projection: about 7.5% in the first 10k steps, dropping below 1% in later phases, which supports the bounded-drift assumption.
>
> **Q3 (Effect of fixing $\gamma = 1$; results for smaller $\gamma$.)**
> We have added a sensitivity study in Appendix F.3 (Table “Sensitivity of PSPO to the discount factor $\gamma$”), sweeping $\gamma \in \{1.0, 0.99, 0.95\}$. In our single-terminal-reward, finite-horizon RLHF setting, $\gamma = 1$ performs slightly best; smaller $\gamma$ mildly weakens temporal credit assignment and yields 0.3–1.1 point lower accuracy on GSM8K/MATH, but training remains stable. We therefore keep $\gamma = 1$ as default and now state that $\gamma$ is a task-dependent hyperparameter.
>
> **Additional concern: strength of theoretical assumptions and reward miscalibration.**
> We address the bounded-drift assumptions and mutual stability in the General Response (items (2)–(3)), including formal propositions and empirical $\delta_k$ diagnostics. To address reward-model miscalibration, Appendix I adds a synthetic stress test with length-biased and saturated rewards, and Section 3.1 and the Limitations clarify that PSPO densifies the given scalar reward but does not correct a fundamentally flawed reward model.
>
> We are grateful for these detailed questions, which helped us clarify both the theoretical assumptions and the practical design choices in PSPO. Thank you again for the careful reading and constructive feedback.

---

### Author Response · Authors · 2025-12-02
**Clarifications, Theoretical Guarantees, and Extended Experiments**

**General Response (to all reviewers)**

We thank all reviewers for their careful reading and constructive feedback. We revised the paper along four main axes: clarifying the method, tightening the theory, adding drift diagnostics, and broadening experiments. Below we briefly summarise the main changes.

1. **Clearer objectives, notation, and workflow.**

   We moved the full training objectives into the main text. Section 3.5 now explicitly defines the shaped reward
   $r^\prime_t$, RTG $G_t$, shaped advantage $A_t^{\text{shape}}$, and the corresponding unshaped $\hat G_t, \hat A_t$, and Section 3.6 plus Algorithm 1 give a complete description of the alternating policy–potential workflow.

2. **Theory: mutual stability, policy invariance, and trust region.**

   Section 4 and Appendix A formalise “mutual stability” (bounded cumulative drift of $\Phi$ and a KL trust region on the policy), restate the potential-based policy-invariance result of Ng et al. (1999) for time-varying potentials, and verify the assumptions for AdamW+cosine decay and an $\ell_\infty$ trust region. We also introduce an explicit regulariser $\mathcal{L_TR}$ and optional projection on $\|\Phi_{\theta_{k+1}} - \Phi_{\theta_k}\|_\infty$.

3. **Drift diagnostics and target choice for $\Phi_\theta$.**

   Appendix H.1 now reports empirical statistics of successive potential change $\delta_k$ and the fraction of batches that trigger the projection, showing that drift decays quickly and projections are rare after early training. Section 3.5 and Appendix A.1 also justify training $\Phi_\theta$ on unshaped, centred advantages $\hat A_t$ via a bias–variance argument, and we add a small ablation showing that using raw returns degrades performance and increases variance.

4. **Hyperparameters and reward miscalibration.**

   Appendix F.3 reports a sweep over $\gamma \in \{1.0, 0.99, 0.95\}$; in our single-terminal-reward setting $\gamma = 1$ performs slightly best, so we keep it as default and note it is tunable. Appendix I adds a synthetic miscalibration stress test (length-biased and saturated rewards), showing that while both GRPO and PSPO degrade, PSPO maintains a margin and does not collapse to reward hacking; we explicitly clarify that PSPO densifies but does not repair a biased reward model.

5. **Internal signals, data, and convergence.**

   Appendix H.2–H.3 provide correlation and control experiments (External-only, Random, Shuffled diagnostics), indicating that internal signals carry non-trivial progress information beyond length. Section 5.1 and Appendix D detail the public math RLHF mixture, clarify that we start from the Qwen2.5-14B base model and use a fixed 300M-token budget, and add a baseline sanity check plus learning curves and longer-budget runs; PSPO consistently stays ahead of PPO/DPO/GRPO, indicating both faster convergence and a modest asymptotic gain in this regime.

6. **Relation to dense/shaping baselines and scope.**

   Appendix K and Appendix L add head-to-head comparisons with LM-Critic, Dense Reward for Free, PAR, and on-policy distillation (GKD) on the same backbone, including VRAM and wall-clock, showing that PSPO is competitive or better while remaining critic-free and lightweight. We also report additional instruction-following results (ShareGPT, HelpfulQA, MT-Bench, AlpacaEval 2.0) and experiments on a Qwen3-4B reasoning model, and we rewrite Section 2.5 and the Limitations to position PSPO as a practical tool for long-horizon, sparse-reward settings with informative internal signals, rather than a universal solution.

---

### Meta-Review · Area_Chair_VfVq · 2026-01-06

**Summary:**

The reviewers have concerns about:
1. writing quality and potential heavy usage of AI.
2. strong assumptions in the guarantee
3. Limited baselines for comparison
4. Insufficient training
These concerns undermine the contributions of this work to the field.

**Reviewer Concerns:**

Most of the concerns are resolved. However, the concerns about empirical evaluation and training still remain outstanding.

**Reviewer Scores:**

4 4 4 4

---

### Decision · Program_Chairs · 2026-01-26

Reject